# Na$^+$/K$^+$ pump interacts with the *h*-current to control bursting activity in central pattern generator neurons of leeches

Daniel Kueh[1], William H Barnett[2], Gennady S Cymbalyuk[2], Ronald L Calabrese[1]*

[1]Department of Biology, Emory University, Atlanta, United States; [2]Neuroscience Institute, Georgia State University, Atlanta, United States

**Abstract** The dynamics of different ionic currents shape the bursting activity of neurons and networks that control motor output. Despite being ubiquitous in all animal cells, the contribution of the Na$^+$/K$^+$ pump current to such bursting activity has not been well studied. We used monensin, a Na$^+$/H$^+$ antiporter, to examine the role of the pump on the bursting activity of oscillator heart interneurons in leeches. When we stimulated the pump with monensin, the period of these neurons decreased significantly, an effect that was prevented or reversed when the *h*-current was blocked by Cs$^+$. The decreased period could also occur if the pump was inhibited with strophanthidin or K$^+$-free saline. Our monensin results were reproduced in model, which explains the pump's contributions to bursting activity based on Na$^+$ dynamics. Our results indicate that a dynamically oscillating pump current that interacts with the *h*-current can regulate the bursting activity of neurons and networks.

*For correspondence: ronald.
calabrese@emory.edu

## Introduction

Rhythmic behaviors such as walking, breathing, and running are controlled by central pattern generators, networks of neurons that produce rhythmic activity without sensory input (*Marder and Calabrese, 1996*; *Marder and Bucher, 2001*). The rhythmic bursting activity of each constituent neuron within a central pattern generator is shaped by the dynamics of various ionic currents that are intrinsic to each neuron (*Harris-Warrick, 2002*). Many of these neurons share subsets of ionic currents with similar functional properties that give rise to bursting activity. For example, many central pattern generator neurons have a persistent Na$^+$ current or a low-threshold Ca$^{2+}$ current that supports bursting (*Opdyke and Calabrese, 1994*; *Butera et al., 1999*; *Del Negro et al., 2002*; *Rybak et al., 2004*), a hyperpolarization-activated inward current that provides recovery from inhibition to initiate bursting (*Angstadt and Calabrese, 1989*; *Golowasch and Marder, 1992*), and a transient K$^+$ current that impedes initiations of action potentials and bursts (*Simon et al., 1992*). Moreover, modulation of these ionic currents can alter the timing and intensity of these neurons' bursting activity (e.g., *Tobin and Calabrese, 2005*; *Koizumi and Smith, 2008*).

Although the Na$^+$/K$^+$ pump is ubiquitous in all animal cells, its role in regulating the bursting activity of neurons in general has not been widely considered. The pump is a transmembrane protein that maintains the intracellular concentrations of Na$^+$ and K$^+$ by exchanging three intracellular Na$^+$ ions for two extracellular K$^+$ ions with each cycle of ATP (adenosine triphosphate) hydrolysis (*Thomas, 1972a*; *De Weer and Geduldig, 1973*). Because the exchange of Na$^+$ and K$^+$ ions is unequal, the pump is electrogenic as it generates an outward current (*Glitsch, 2001*). In addition to maintaining internal concentrations of Na$^+$ and K$^+$, the pump contributes a voltage drop to the resting membrane potential (*Hodgkin and Keynes, 1955*; *Carpenter and Alving, 1968*; *Smith et al., 1968*; *Baylor and Nicholls, 1969*) and is able to generate a slow afterhyperpolarization after a train

**eLife digest** In animals, cells called neurons relay information around the body in the form of electrical signals. An enzyme called the sodium and potassium pump is found in the membrane that surrounds neurons. It uses energy to pump sodium ions out of the neuron and potassium ions in the opposite direction. This helps to maintain different concentrations of these ions across the membrane, which is critical for the electrical activity of neurons and also generates an electrical current in the process. The size of the current is influenced by how many sodium ions have leaked back into the neuron due to the neuron's electrical activity.

Neurons control many rhythmic processes in animals including breathing and heartbeats. However, it was not clear how the current produced by the sodium and potassium pump contributes to the rhythms in neural activity that drive these processes. To address this question, Kueh et al. investigated the effect of drugs that alter the activity of the pump in neurons that control heartbeat in leeches.

The experiments show that stimulating the pump by altering the amount of sodium ions that leak into the neuron dramatically sped up the rhythmic activity of these neurons. This effect depended completely on the presence of a channel protein – called an *h*-channel – that was activated with a delay by the altered pump current and allowed sodium and potassium ions to cross the membrane, counteracting the pump current. Inhibiting the pump also sped up the rhythm of neural activity, but this effect did not depend on the *h*-channel. Kueh et al. developed a computer model that indicated that the time course of the pump current following the sodium ion leak and the slow activation of the *h*-channel were important for these effects.

Previous studies have shown that a particular signal molecule modulates the activity of both the pump and the *h*-channel in neurons. Therefore, a future challenge is to find out how the pump and the *h*-channel interact while their activities change in response to the signal molecule.

of action potentials when its activity is enhanced by increased intracellular Na$^+$ (*Gage and Hubbard, 1966*; *Nakajima and Takahashi, 1966*; *Rang and Ritchie, 1968*; *Baylor and Nicolls, 1969*; *Sokolove and Cooke, 1971*; *Bolton, 1973*; *Mat Jais et al., 1986*; *Gordon et al., 1990*; *Catarsi and Brunelli, 1991*; *Lombardo et al., 2004*; *Pulver and Griffith, 2010*; *Gulledge et al., 2013*). In the context of motor patterns, the pump appears to play an important role in regulating bursting activity (*Ballerini et al., 1997*; *Tobin and Calabrese, 2005*; *Zhang and Sillar, 2012*). For example, *Zhang and Sillar (2012)* found that blocking the pump with ouabain abolished the slow afterhyperpolarization of spinal cord central pattern generator neurons in *Xenopus laevis* tadpoles, resulting in longer swimming episodes. In a separate study, *Tobin and Calabrese (2005)* observed that inhibition of the pump with the neuropeptide, myomodulin, or with ouabain speeds up the bursting activity of oscillator heart interneurons in the leech heartbeat central pattern generator (*Tobin and Calabrese, 2005*). These studies show that the pump can serve as a target for modulating the bursting activity of neurons and networks that program motor output.

Many studies have explored the function of the pump by inhibiting its activity; fewer have investigated the pump's function by stimulating its activity. For example, *Zhang et al. (2015)* recently found that stimulating the pump activity of central pattern generator neurons in *Xenopus* enhances the ultraslow hyperpolarization, which suppresses excitability of the entire motor network. Nevertheless, the effects of stimulating pump on the ongoing activity of rhythm generating neurons have yet to be explored. In the present study, we had two principal goals. First, we wanted to reveal experimentally the mechanisms that underlie the effects of a stimulated pump on the bursting activity of central pattern generator neurons, especially with respect to the temporal or burst characteristics of these neuron's bursting activity such as period or duty cycle. Second, we wanted to develop a mathematical model that could capture our experimental results and help identify mechanisms. For our analysis, we used leech oscillator heart interneurons, which participate in half-center oscillators to pace the heartbeat central pattern generator. We examined the influence of the pump on bursting based on changes in the burst characteristics of these oscillator heart interneurons. We used monensin, a Na$^+$/H$^+$ antiporter, to increase Na$^+$ concentrations to stimulate the pump (*Hill and Licis,*

*1982*). Our results show that monensin enhances the outward pump current, which hyperpolarizes the membrane potential of oscillator heart interneurons. We also found that stimulation of pump activity by monensin speeds up the bursting activity of oscillator heart interneurons. Blocking the *h*-current of these neurons with $Cs^+$ while stimulating the pump with monensin failed to speed up bursting. Our biophysical model captured these experimental results by simulating the interaction between the pump current and the *h*-current to control the interburst interval and thus the period. Taken together, our study leads us to conclude that in the presence of the *h*-current, the electrogenic activity of the pump can play a significant role in the dynamics of bursting activity in the leech heartbeat central pattern generator and likely in other rhythmically bursting neuronal networks.

## Results

### Intracellular leakage of $Na^+$ hyperpolarizes the membrane potential and suppresses the spiking activity of oscillator heart interneurons

To delineate the role of the $Na^+/K^+$ pump in central pattern generator neurons, we used oscillator heart interneurons that pace bursting activity in the leech heartbeat central pattern generator. There are two pairs of these oscillator heart interneurons in each animal, with each pair located in the third and fourth segmental ganglia of the ventral nerve cord. Both neurons in each ganglion form mutual inhibitory synaptic connections, thereby constituting a half-center oscillator. We used only individual isolated ganglia in our experiments to determine the contribution of the pump to ongoing rhythmic bursting in oscillator heart interneurons. The pump current is proportional to the pump rate, which is itself proportional to intracellular $Na^+$ concentrations (*Thomas, 1972a*). Thus, the pump can be stimulated by increasing the intracellular loading of $Na^+$ from an electrode filled with a $Na^+$-based solution, which has been demonstrated in other neurons such as mechanoreceptors in leeches (*Jansen and Nicholls, 1973*; *Catarsi and Brunelli, 1991*; *Lombardo et al, 2004*) and snails (*Kerkut and Thomas, 1965*).

To determine if the intracellular loading of $Na^+$ hyperpolarizes the membrane potential of the oscillator heart interneurons, we recorded one oscillator heart interneuron with an extracellular electrode and impaled the contralateral oscillator heart interneuron with an intracellular electrode, filled with the standard 2M KAcetate and 20 mM KCl solution, in normal saline. Upon impalement, we measured the base potential, defined as midway between an undershoot trough and the next threshold (first peak of the third derivative) of the intracellularly recorded neuron in the first 10 min. Within 3 min of impalement, the base potential stabilized to $-40.8 \pm 1.6$ mV (n = 5), a voltage that is consistent with previous observations (*Olsen and Calabrese, 1996*), and both neurons exhibit their usual rhythmic bursts of action potentials for 20 min or more (*Figure 1A$_1$*). In another group of preparations, we substituted the KAcetate and KCl with equimolar concentrations of NaAcetate and NaCl and recorded the activity of the oscillator heart interneurons in normal saline. When an oscillator heart interneuron was impaled with a $Na^+$-filled electrode, its rhythmic activity decreased rapidly over time and its base potential was noticeably more hyperpolarized than the $K^+$-loaded neurons (*Figure 1A$_2$*). We compared the average base potential of both groups of preparations for the first ten minutes and found that the base potential of the $Na^+$-loaded neurons was significantly more hyperpolarized than the base potential of the $K^+$-loaded neurons (*Figure 1A$_3$*, n = 5, split-plot ANOVA, $F_{1,8}$ = 1847.7, p=0.006). Despite being hyperpolarized, the $Na^+$-loaded neurons were very responsive to brief depolarizing pulses, indicating that they were still healthy. In summary, these results show that the intracellular leakage of $Na^+$ does hyperpolarize the membrane potential of oscillator heart interneurons.

### Monensin suppresses spiking activity and hyperpolarizes the membrane potential of oscillator heart interneurons when the *h*-current and all $Ca^{2+}$ currents are blocked

To determine the contributions of the $Na^+/K^+$ pump under multiple experimental treatments, we used monensin, an antibiotic that functions as a $Na^+$-$H^+$ antiporter in cell membranes (*Lichtshtein et al., 1979*; *Hill and Licis, 1982*; *Zhang et al., 2015*). Monensin has been found to increase intracellular $Na^+$ concentrations, which stimulates the pump (*Lichtshtein et al., 1979*; *Hill and Licis, 1982*; *Zhang et al., 2015*) and hyperpolarizes the membrane potential of various cell

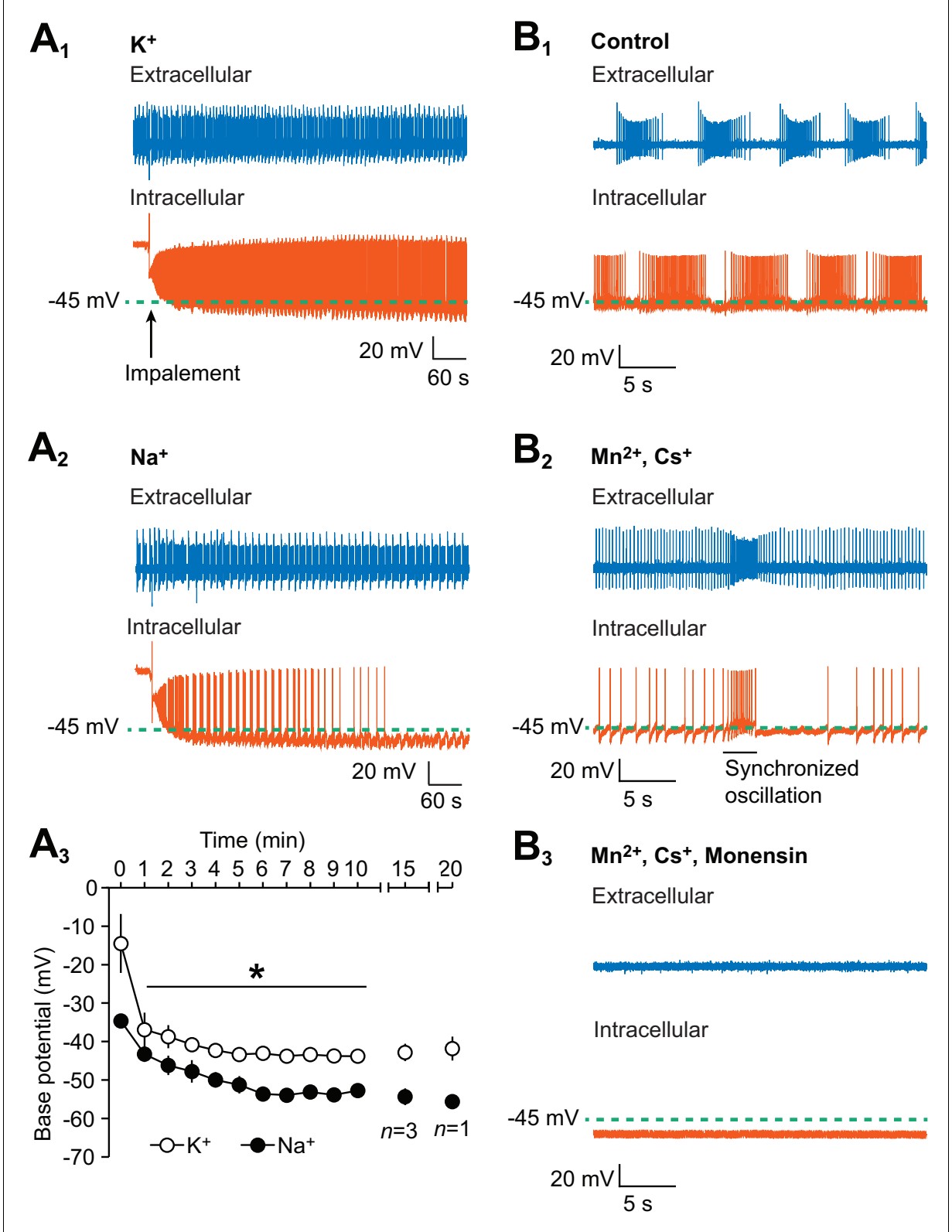

**Figure 1.** Hyperpolarization of the oscillator heart interneurons and suppression of their spiking activity by intracellular leakage of Na$^+$ from an electrode and by monensin. (A$_1$) An extracellular (blue) trace of one oscillator heart interneuron and an intracellular (vermilion) trace of a contralateral oscillator heart interneuron that was impaled with a K$^+$-filled intracellular electrode. (A$_2$) Impalement of an oscillator heart interneuron with a Na$^+$-filled electrode gradually suppressed its spiking activity and hyperpolarized the neuron. There was no change in the bursting activity of the extracellularly

*Figure 1 continued on next page*

Figure 1 continued

recorded neurons in the ($A_1$) $K^+$ and ($A_2$) $Na^+$ recordings. ($A_3$) During the first ten minutes, the average base potential of $Na^+$-loaded neurons (closed circles) was significantly more hyperpolarized than the base potential of $K^+$-loaded neurons (open circles). Such differences persisted well into the 15th and 20th minute. The data are represented as mean ± SEM, with the asterisk (*) representing significant differences between the $K^+$ and $Na^+$ base potentials (split-plot ANOVA, $F_{1,8} = 1847.7$, p=0.006). ($B_1$) Extracellular (blue) and intracellular (vermilion) traces from a pair of oscillator heart interneurons that were initially bathed in control saline and showed normal alternating bursting. ($B_2$) When the oscillator heart interneurons were bathed in $Ca^{2+}$-free saline with 2 mM $Cs^+$ and 1.8 mM $Mn^{2+}$, they produced a more tonic firing pattern that was interspersed with synchronized oscillations. ($B_3$) When the oscillator heart interneurons were subsequently treated with 10 µM monensin in the same $Ca^{2+}$-free saline, the spiking activity of both oscillator heart interneurons were suppressed and the membrane potential of the intracellularly recoded neuron gradually hyperpolarized.

types from other systems (*Lichtshtein et al., 1979*; *Tsuchida and Otomo, 1990*; *Satoh and Tsuchida, 1999*; *Doebler, 2000*; *Wang et al., 2012*). To determine if monensin also hyperpolarizes the membrane potential of oscillator heart interneurons from the leech heartbeat system, we performed dual intracellular and extracellular recordings from a pair of oscillator heart interneurons and measured the base potential of the intracellularly recorded neuron before and after external application of monensin.

In normal saline, both oscillator heart interneurons fired their usual alternating bursts of action potentials and the base potential of the intracellularly recorded neuron was $-41.5 \pm 1.5$ mV (n = 5). We then applied $Ca^{2+}$-free saline, which contained 2 mM $Cs^+$ to block the $h$-current (*Angstadt and Calabrese, 1989*, *1991*) and 1.8 mM $Mn^{2+}$ to block all $Ca^{2+}$ currents as well as synaptic transmission (*Angstadt and Calabrese, 1991*). Once the $Ca^{2+}$-free saline took effect, the oscillator heart interneurons no longer burst regularly but instead fired tonically, indicating that the neurons were synaptically isolated and incapable of normal bursting (*Figure 1B_2*). Such tonic firing was interspersed by brief $Na^+$-based synchronized oscillations (*Figure 1B_2*), a characteristic of $Mn^{2+}$ exposure in leech neurons (*Angstadt and Friesen, 1991*). We measured the base potential of these neurons after these synchronized oscillations had appeared, which was 2–7 min after applying the $Ca^{2+}$-free saline. We then added 10 µM monensin to the $Ca^{2+}$-free saline, which abolished the synchronized oscillations and eventually suppressed spiking activity (*Figure 1B_3*). We chose the concentration of 10 µM for monensin because we wanted to produce the maximum effect within the shortest amount of time (see details below on the effects of lower concentrations of monensin on bursting activity). Within 7–10 min of adding monensin, the base potential under the $Ca^{2+}$-free monensin saline became significantly more hyperpolarized than the base potential under $Ca^{2+}$-free saline (*Figure 1B_3*, $-43 \pm 1.8$ mV for pre-monensin vs. $-54.4 \pm 3.4$ mV for monensin, n = 5, paired t-test, p=0.02). Moreover, the suppression of spiking activity in the extracellularly recorded neurons is consistent with the hyperpolarization of the intracellularly recorded neurons (*Figure 1B_3*). Thus, consistent with our hypothesis and with previous studies (e.g., *Doebler, 2000*; *Lichtshtein et al, 1979*; *Tsuchida and Otomo, 1990*; *Satoh and Tsuchida, 1999*; *Wang et al., 2012*), monensin hyperpolarizes the membrane potential of oscillator heart interneurons, which suppresses their spiking activity.

## Monensin enhances the outward current generated by the $Na^+$-$K^+$ pump in oscillator heart interneurons

To determine if the hyperpolarized membrane potential by monensin was due to the outward current generated by a stimulated $Na^+$/$K^+$ pump, we voltage-clamped one of the oscillator heart interneurons from an isolated ganglion and looked for shifts in the membrane current brought about by monensin as well as by monensin plus strophanthidin (*Figure 2A_1*). If monensin does indeed stimulate the pump, we should see an outward shift in the membrane current. We first bathed the oscillator heart interneurons in the same $Ca^{2+}$-free saline to block the $h$-current and all $Ca^{2+}$ currents, thereby also suppressing synaptic transmission. The neurons were then voltage-clamped at $-45$ mV for five minutes and later treated with 10 µM monensin for ten minutes (*Figure 2A_1*). The holding potential of $-45$ mV was chosen because it is near the observed base potential of oscillator heart interneurons when bathed in $Ca^{2+}$-free saline (*Figure 1B_2*). Moreover, we wanted to reduce the escape spiking of these neurons (*Norris et al., 2007*). Before adding monensin, the average holding current at $-45$ mV was $-26.4 \pm 16.9$ pA (*Figure 2A_2*, n = 5), which was measured at the fifth minute

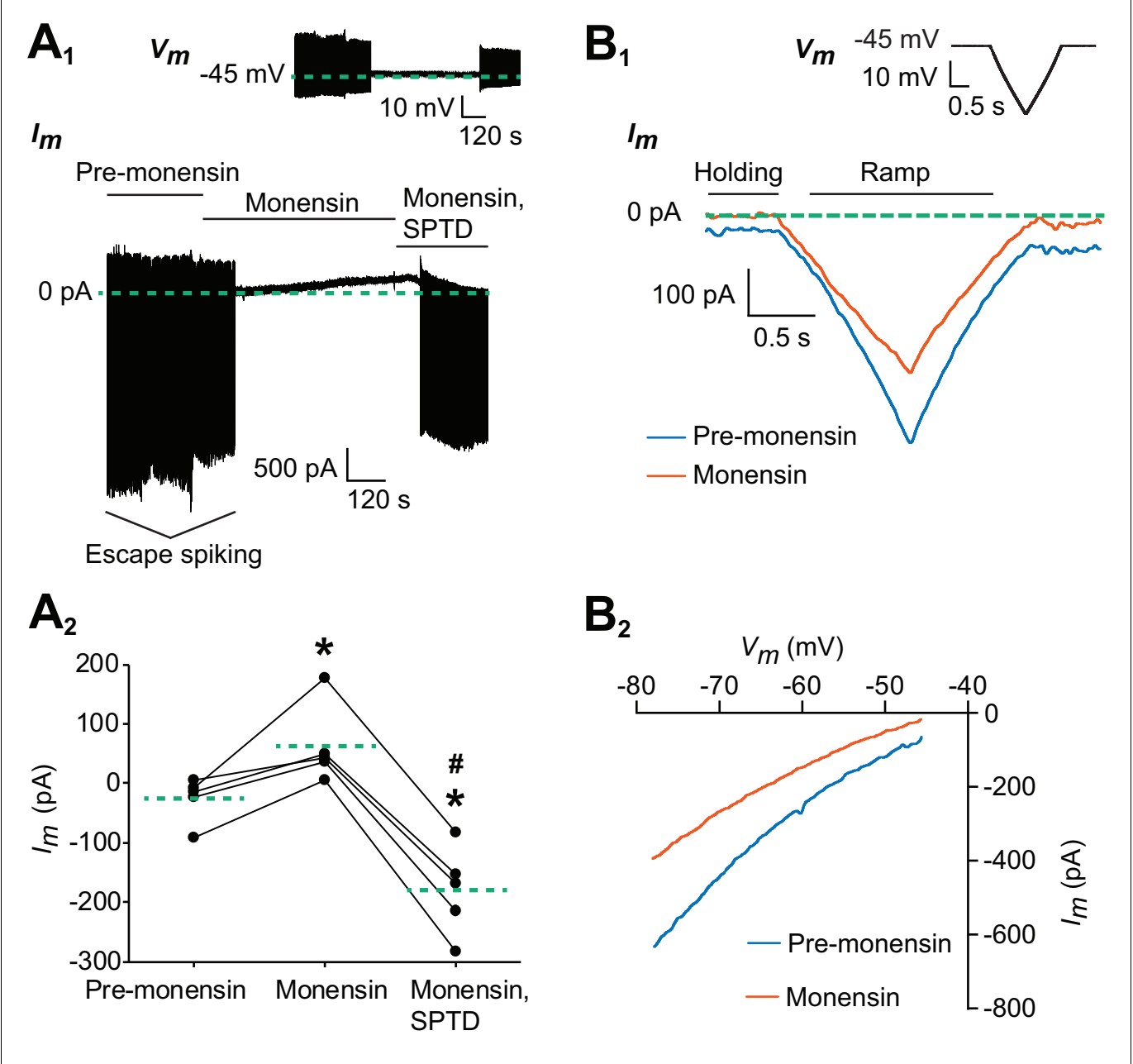

**Figure 2.** Monensin stimulates the outward $Na^+/K^+$ pump current. ($A_1$) Membrane current trace from an oscillator heart interneuron with its membrane potential ($V_m$) voltage-clamped at −45 mV (see inset) in $Ca^{2+}$-free saline with 1.8 mM $Mn^{2+}$ plus 2 mM Cs. Changes in the neuron's membrane current ($I_m$) were observed under three experimental treatments: pre-monensin saline for five minutes, 10 μM monensin for 10 min, and 10 μM monensin plus 100 μM strophanthidin (SPTD) for another five minutes. ($A_2$) A scatterplot of membrane currents from five preparations, with each green dashed line representing a mean for each of the three experimental treatments. Monensin induced a significant outward current relative to pre-monensin saline. Monensin plus strophanthidin induced a significant inward current relative to pre-monensin or monensin saline. The asterisks (*) represent significance from the pre-monensin saline whereas the hashtag (#) represents significance from the monensin saline (Tukey's test, p<0.05 for all tests). ($B_1$) Membrane currents from the same oscillator heart interneuron that was voltage-clamped at −45 mV in the same $Ca^{2+}$-free saline before and after treatment with 10 μM monensin. Both currents were generated by a slow ramp-clamp protocol, with each voltage ramp running from −45 mV to −80 mV and back (see inset). Compared to the membrane current under pre-monensin saline (blue trace), the membrane current under monensin saline (vermilion trace) shifted outwards across the entire range of negative voltage-ramp values. ($B_2$) Current-voltage relations under pre-monensin (blue line) and monensin (vermilion line) treatments from the same oscillator heart interneuron were generated based on the ($B_1$) voltage-ramp traces.

of voltage-clamping the neurons in $Ca^{2+}$-free saline. Monensin suppressed all escape spiking and induced an outward current which moved the holding current to 62.4 ± 29.6 pA (*Figure 2A$_{1-2}$*, n = 5, Tukey's test, p=0.009), measured at the tenth minute after monensin application. This monensin-induced outward current was apparently blocked, however, by the application of 100 μM strophanthidin, resulting in a large inward holding current (*Figure 2A$_{1-2}$*, −180.0 ± 33.3 pA, n = 5, Tukey's test, p=0.0002) and a resumption of escape spiking by the neurons (*Figure 2A$_2$*). The difference between the average outward current generated by monensin and the average inward current observed upon blocking the pump with strophanthidin, suggests that the pump is able to generate a maximum outward current of 242.4 ± 16.0 pA, assuming a saturating effect of 10 μM monensin on the pump. Subtracting the outward current generated by monensin from this maximum pump current, suggests that the resting pump current is about 180.0 ± 33.3 pA (or 74.0 ± 11.2% of the maximum pump current). Thus, the maximum pump current appears to have a limited range of 202 to 289 pA, with about three-quarters of this current being generated under resting conditions, leaving another quarter of the current available for enhancement by 10 μM monensin.

To determine if the outward current produced by monensin is due to an enhanced pump current and not by changes in membrane conductance, we voltage-clamped another set of oscillator heart interneurons in the same $Ca^{2+}$-free saline for 4–6 min, before generating a series of three voltage-ramps using a slow ramp-clamp protocol, with each ramp running from −45 mV to −80 mV and back (See inset in *Figure 2B$_1$*). We then added 10 μM monensin to the $Ca^{2+}$-free saline for 6–9 min before generating another series of voltage ramps (*Figure 2B$_1$*). In four of the five preparations, we saw outward shifts in the membrane current across the entire range of negative voltage-ramp values (*Figure 2B$_1$*). These outward shifts occurred without any significant change in conductance (*Figure 2B$_2$*, 15.3 ± 1.6 nS for pre-monensin saline vs. 13.0 ± 1.3 nS for monensin saline, n = 5), which is consistent with our hypothesis that the outward current brought induced by monensin is driven by stimulation of the pump and not by changes in conductance.

## Stimulation of the $Na^+$/$K^+$ pump with monensin speeds up the bursting activity of oscillator heart interneurons

Since we have established that monensin stimulates the $Na^+$/$K^+$ pump, which hyperpolarizes the membrane potential, we predicted that treating the oscillator heart interneurons with monensin would slow down their bursting activity as indicated by an increased period (for the definition of period, see *Figure 3A* and Materials and methods). To test this hypothesis, we again performed dual intracellular and extracellular recordings from a pair of oscillator heart interneurons in normal saline. Contrary to what we expected, when we applied 10 μM monensin for 15 min, we found that the average period of both oscillator heart interneurons actually decreased (*Figure 3*, 9.3 ± 1.0 s for control saline vs. 3.9 ± 0.2 s for monensin saline, n = 5, paired t-test, p=0.007), which were detectable within two minutes of applying monensin. We measured the period after 5.1–10.5 min of adding monensin because shortening of the period stabilized during this time period (see example in *Figure 3—figure supplement 1*). In four of the five preparations, the activities of both neurons initially transitioned from bursting to tonic-like firing and back to bursting again (*Figure 3B*). The disorganized tonic-like firing pattern did not appear in the extracellular recordings (*Figure 4*), leaving us to conclude that this pattern was due to a nonspecific leak current introduced by the intracellular electrode (*Cymbalyuk et al., 2002*). We also measured the base potential before and after application of monensin but did not find any significant changes. Thus, monensin actually speeds up the bursting activity of oscillator heart interneurons without significantly affecting their base potential.

Because it was difficult to sustain a healthy intracellular recording from an oscillator heart interneuron for more than 40 min, we were not able to determine if the effects of monensin are reversible in our intracellularly-recorded cells beyond 15–20 min of washout with normal saline. To determine if the effects of monensin could be reversed with longer washouts, we recorded extracellularly from both oscillator heart interneurons in normal saline to overcome the difficulties of obtaining long-term recordings and to prevent disorganized firing patterns from being introduced by intracellular electrodes. We treated the oscillator heart interneurons with 10 μM monensin for 10 min, followed by a washout with normal saline for six hours. There was variability across preparations with respect to when the period started to increase, which occurred between 20.2 to 76.9 min. In preparations that were exposed to monensin for 20–50 min, we found that the effects of 10 μM monensin are not reversible.

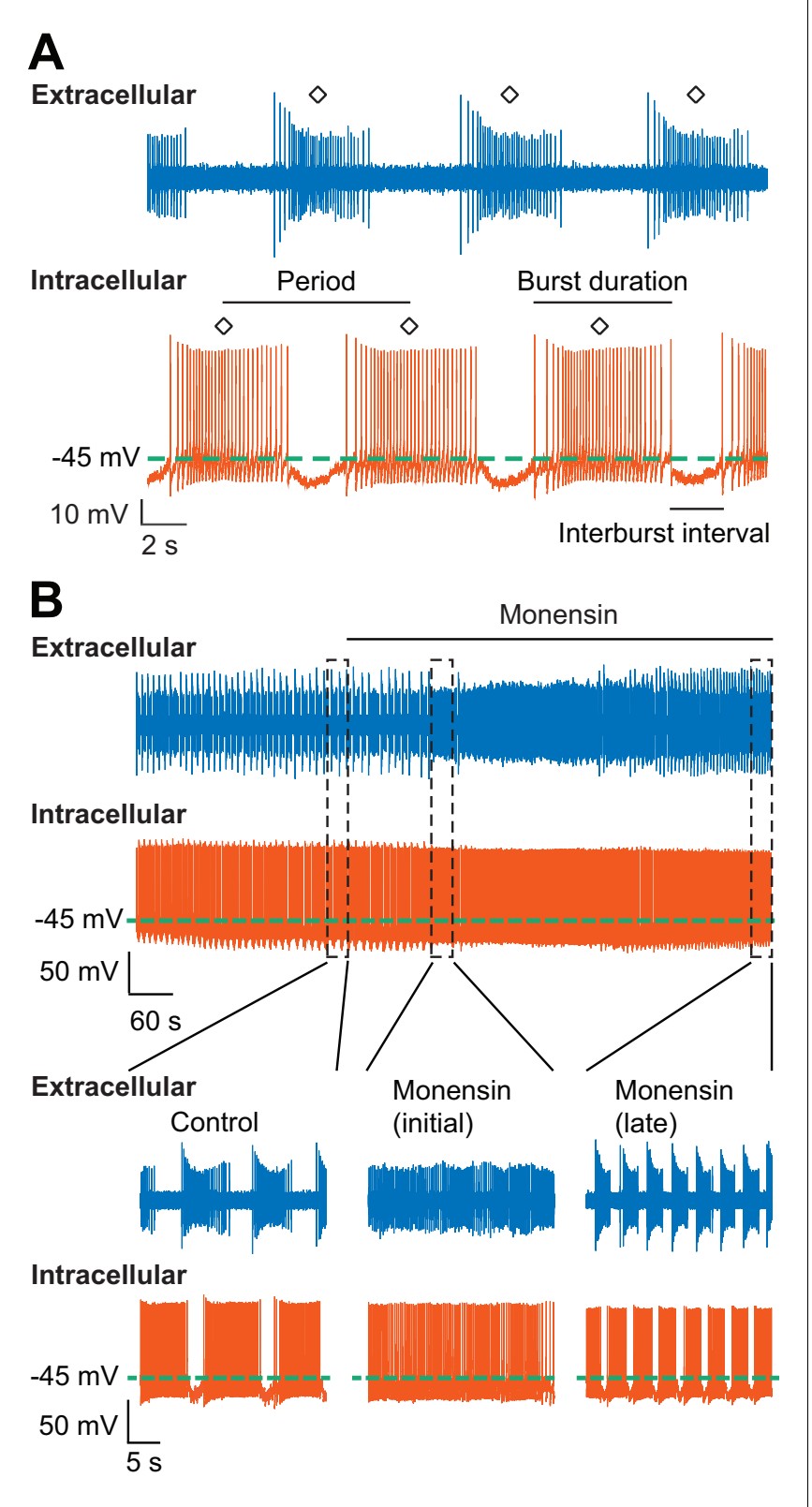

**Figure 3.** Stimulation of the Na⁺-K⁺ pump with monensin speeds up the bursting activity of oscillator heart interneurons as half-center oscillators. (**A**) Extracellular (blue) and intracellular (vermilion) traces from a pair of oscillator heart interneurons functioning as a half-center oscillator. Burst characteristics such as the period, burst duration, and interburst interval can be measured from each neuron's bursting activity. The period was measured

*Figure 3 continued on next page*

*Figure 3 continued*

from the middle action potential (diamond symbol) of one burst to the middle action potential of the next burst. (B) Extracellular (blue) and intracellular (vermilion) traces from a pair of oscillator heart interneurons. In control saline, both neurons function as a half-center oscillator by firing alternating bursts of action potentials. Adding 10 μM monensin to the saline resulted in initial tonic firing by both neurons followed by alternating bursts of action potentials with a reduced period.

The following figure supplement is available for figure 3:

**Figure supplement 1.** The cycle-to-cycle effects of monensin on the period of oscillator heart interneurons.

To determine if the decreased period observed in our combined intracellular and extracellular experiments with monensin (*Figure 3B*) was due to the 10 μM concentration that we used, we performed another set of dual extracellular recordings and treated the oscillator heart interneurons with 1 μM monensin for 30 min followed by the treatment of 10 μM monensin for another 15 min. When we increased the concentration of monensin from 1 to 10 μM, only the burst duration decreased significantly, but only by just $0.7 \pm 0.1$ s ($2.4 \pm 0.1$ s for 1 μM monensin vs. $1.7 \pm 0.1$ s for 10 μM monensin saline, n = 5, Tukey's test, p=0.044). Thus, increasing the concentration of monensin in the saline from 1 to 10 μM does not appear to affect the period of neurons that have already been treated with monensin.

To determine if the effects of 1 μM monensin occurs more slowly than 10 μM monensin, we performed another set of extracellular recordings with 10 μM monensin and compared the amount of time needed for the period to reach its minimum value at the 200th cycle in both 1 μM and 10 μM monensin (*Figure 3—figure supplement 1A*). It took significantly longer for the period to reach its 200th cycle value in neurons treated with 1 μM monensin relative to neurons treated with 10 μM monensin (*Figure 3—figure supplement 1B*, $18.5 \pm 0.7$ min for 1 μM monensin vs. $14.5 \pm 0.6$ min for 10 μM monensin saline, n = 5, unpaired t-test, p=0.003). Moreover, the period at the 200th cycle for neurons treated with 10 μM monensin was significantly lower than the period of neurons treated with 1 μM monensin (*Figure 3—figure supplement 1C*, $3.9 \pm 0.1$ s for 1 μM monensin vs. $2.9 \pm 0.1$ s for 10 μM monensin saline, n = 5, unpaired t-test, p=0.00003). Thus, in subsequent experiments, we opted to use 10 μM monensin because its maximal effects can be observed sooner. Moreover, the 10 μM concentration is widely used (e.g., *Lichtshtein et al., 1979*; *Tsuchida and Otomo, 1990*; *Busciglio et al., 1993*; *Itoh et al., 2000*; *Lamy and Chatton, 2011*; *Wang et al., 2012*; *Zhang et al., 2015*), allowing us to compare our results to those from other studies.

## Stimulating the Na⁺/K⁺ pump with monensin while blocking the *h*-current increases the period and interburst interval

We next sought to determine if stimulation of the Na⁺/K⁺ pump by monensin might exert its effect on the period through activation of the *h*-current. Previous studies in other systems have suggested that the *h*-current might counteract the hyperpolarizing effects of the pump current (e.g., *Robert and Jirounek, 1998*; *Soleng et al., 2003*; *Baginskas et al., 2009*). Moreover, *Hill et al. (2001)* developed a canonical model of the oscillator heart interneurons and showed that increasing the *h*-current conductance decreases the period, which was later supported by dynamic clamp experiments (*Sorensen et al., 2004*; *Olypher et al., 2006*). Using a hybrid half-center oscillator, *Sorensen et al. (2004)* found that increasing the *h*-current conductance of living or silicon neurons with dynamic clamp decreases the period and interburst interval, without affecting the burst duration. *Angstadt and Calabrese (1989)* found that the *h*-current in leech oscillator heart interneurons can be blocked by 2–4 mM external Cs⁺. Thus, if we were to block the *h*-current with Cs⁺ while stimulating the pump with monensin, we should observe an increase in the period in oscillator heart interneurons.

To determine if the pump current interacts with the *h*-current to regulate the period and other burst characteristics of oscillator heart interneurons, we performed dual extracellular recordings as previously described and added 10 μM monensin to the saline for 15 min followed by the application of 10 μM monensin plus 2 mM Cs⁺ to block the *h*-current for another 15 min (*Figure 4A₁₋₃*). We then compared five burst characteristics (see *Figure 3A* for definitions of period, burst duration, and

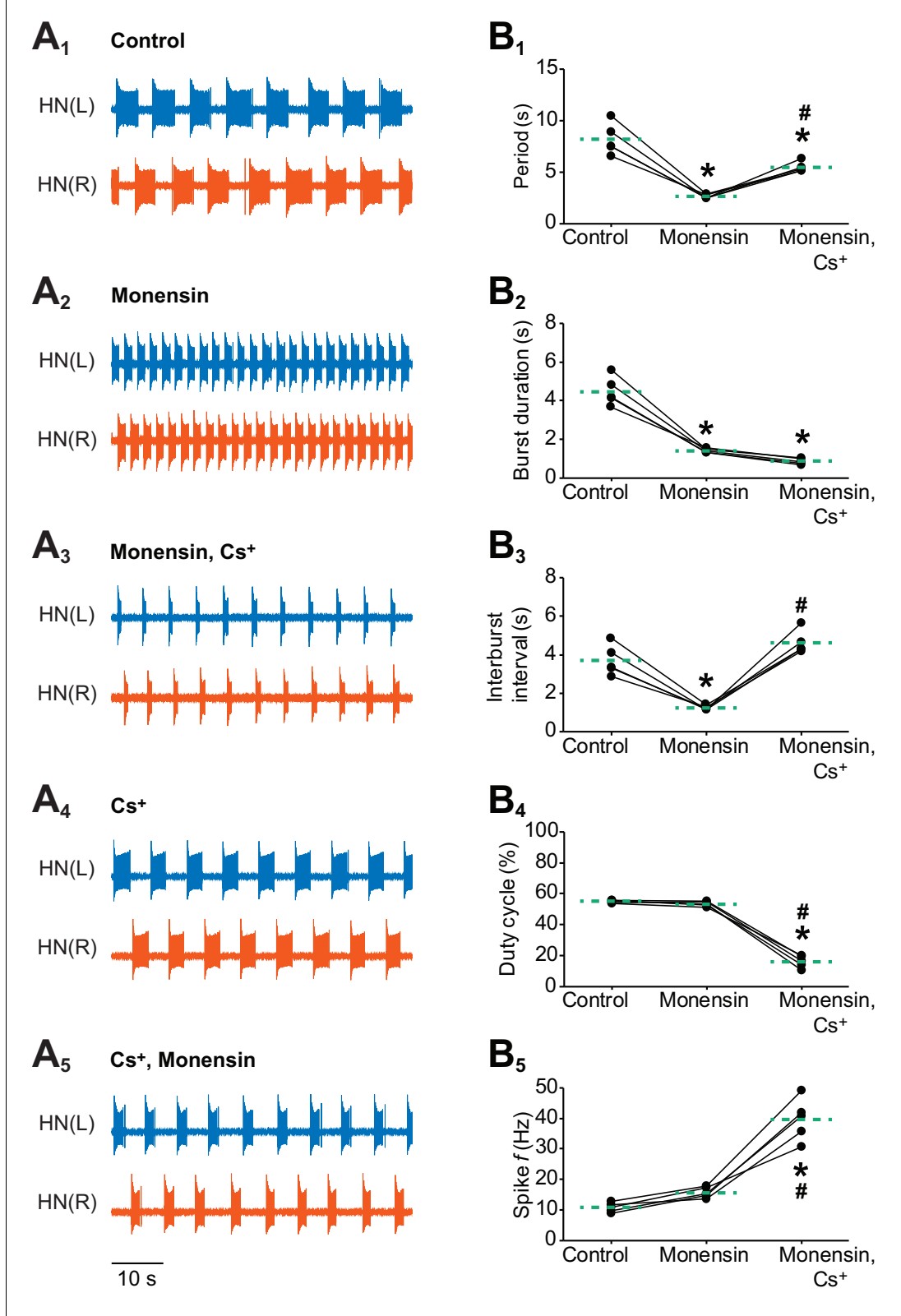

**Figure 4.** Effects of stimulating the Na$^+$-K$^+$ pump with monensin while blocking the *h*-current on the burst characteristics of oscillator heart interneurons as half-center oscillators. Extracellular traces of bursting activity by left (blue) and right (vermilion) oscillator heart interneurons [HN(L) and HN(R) neurons] initially bathed in (A$_1$) control (normal) saline and subsequently treated with (A$_2$) saline that contained 10 μM monensin followed by another treatment with (A$_3$) saline that contained 10 μM monensin plus 2 mM Cs$^+$. Extracellular traces of another pair of oscillator heart interneurons
*Figure 4 continued on next page*

*Figure 4 continued*

first treated with ($A_4$) saline that contained 2 mM $Cs^+$ followed by another treatment with ($A_5$) saline that contained 2 mM $Cs^+$ plus 10 µM monensin. Scatterplots of the ($B_1$) period, ($B_2$) burst duration, ($B_3$) interburst interval, ($B_4$) duty cycle, and ($B_5$) intraburst spike frequency that were measured from the extracellular traces of five preparations under ($A_{1-3}$) three experimental treatments. Monensin decreased significantly the ($B_1$) period, ($B_2$) burst duration, and ($B_3$) interburst interval relative to control. The ($B_4$) duty cycle and ($B_5$) intraburst spike frequency were unchanged. Monensin plus $Cs^+$ increased significantly ($B_1$) the period, ($B_3$) interburst interval, and ($B_5$) intraburst spike frequency relative to control. Because the ($B_2$) burst duration under monensin plus $Cs^+$ remained unchanged relative to monensin alone, the ($B_4$) duty cycle decreased significantly under the monensin plus $Cs^+$ saline relative to either control or monensin saline. The dashed green lines represent means whereas asterisks (*) and hashtags (#) represent significance from control and monensin, respectively (Tukey's test, p<0.05).

The following figure supplement is available for figure 4:

**Figure supplement 1.** Stimulating the pump with 10 µM monensin in pharmacologically isolated heart interneurons requires *h*-current to shorten the interburst interval but not to shorten the burst duration.

interburst interval and Materials and methods for definitions of duty cycle and intraburst spike frequency) across all three experimental treatments (*Figure 4B₁₋₅*). Similar to the previous intracellular and extracellular recordings, we found that monensin decreased the period significantly relative to control saline (*Figure 4A₁₋₂ and B₁*, 8.2 ± 0.7 s for control saline vs. 2.7 ± 0.1 s for monensin saline, n = 5, Tukey's test, p=0.0002). Monensin also significantly decreased the burst duration (*Figure 4A₁₋₂ and B₂*, 4.5 ± 0.3 s for control saline vs. 1.4 ± 0.04 s for monensin saline, n = 5, Tukey's test, p=0.0002) and the interburst interval (*Figure 4A₁₋₂ and B₃*, 3.7 ± 0.3 s for control saline vs. 1.2 ± 0.1 s for monensin saline, n = 5, Tukey's test, p=0.0007), resulting in no significant change in the duty cycle (*Figure 4B₄*). Monensin did not significantly affect the intraburst spike frequency (*Figure 4B₅*). Taken together, stimulation of the pump with monensin speeds up the bursting activity of half-center oscillators without affecting their duty cycle.

When we added $Cs^+$ to the monensin saline, we found that the period increased significantly under monensin plus $Cs^+$ saline relative to the monensin saline (*Figure 4A₂₋₃ and B₁*, 2.7 ± 0.1 s for monensin saline vs. 5.5 ± 0.2 s for monensin plus $Cs^+$ saline, n = 5, Tukey's test, p=0.004). The burst duration remained (*Figure 4A₂₋₃ and B₂*) the same whereas the interburst interval (*Figure 4A₂₋₃ and B₃*, 1.2 ± 0.1 s for monensin vs. 4.6 ± 0.3 s for monensin plus $Cs^+$, n = 5, Tukey's test, p=0.0002) increased significantly relative to monensin alone. As a result, the duty cycle decreased significantly under the monensin plus $Cs^+$ saline relative to monensin saline (*Figure 4B₄*, 53.3 ± 0.7% for monensin saline vs. 16.0 ± 1.8% for monensin plus $Cs^+$ saline, n = 5, Tukey's test, p=0.0002). Finally, the intraburst spike frequency under $Cs^+$ plus monensin saline increased significantly relative to monensin alone (*Figure 4B₅*, 15.7 ± 0.8 Hz for monensin saline vs. 39.6 ± 3.1 Hz for monensin plus $Cs^+$ saline, n = 5, Tukey's test, p=0.0002). Thus, stimulating the pump with monensin decreases the burst duration and the interburst interval. However, its effects on the interburst interval is inhibited when the *h*-current is blocked.

We wished to determine if stimulating the pump with monensin can still exert its effect on the burst duration when the *h*-current has already been blocked, thereby preventing it from affecting the interburst interval and period. We first blocked the *h*-current with $Cs^+$ (*Figure 4A₄*) and then stimulated the pump with 10 µM monensin (*Figure 4A₅*). We found that the burst duration decreased significantly relative to $Cs^+$ saline (4.5 ± 0.7 s for $Cs^+$ saline vs. 2.6 ± 0.4 s for $Cs^+$ plus monensin saline, n = 5, paired t-test, p=0.04). There were no significant changes in the interburst interval (4.8 ± 0.4 s for $Cs^+$ saline vs. 4.4 ± 0.3 s for $Cs^+$ plus monensin saline, n = 5) and period (9.2 ± 1.1 s for $Cs^+$ saline vs. 7.0 ± 0.5 s for $Cs^+$ plus monensin saline, n = 5) relative to $Cs^+$ saline. Thus, when the *h*-current is blocked, stimulating the pump decreases the burst duration but not the interburst interval. This treatment sequence of $Cs^+$ and monensin did not confound our previous results (*Figure 4A₁₋₃*), confirming the critical role of the *h*-current in shortening the interburst interval when the pump is stimulated by monensin.

Similar results were also observed when we stimulated the pump of oscillator heart interneurons that have been isolated with 500 µM bicuculline, which blocks the synaptic transmission between the two oscillator heart interneurons (*Figure 4—figure supplement 1*) (*Schmidt and Calabrese, 1992*; *Cymbalyuk et al., 2002*). Although these oscillator heart interneurons form half-center oscillators,

they can function as endogenous bursters when recorded extracellularly in bicuculline saline (*Cymbalyuk et al., 2002*). The one notable exception was that stimulating the pump with monensin in isolated oscillator heart interneurons does not significantly decrease the period (3.9 ± 0.1 s for bicuculline saline vs. 3.3 ± 0.1 s for bicuculline plus monensin saline, n = 5), which can be explained in part by a floor effect; period had already been significantly reduced in bicuculline saline (8.8 ± 0.9 s for control saline vs. 3.9 ± 0.1 s for bicuculline saline, n = 5, Tukey's test, p<0.001), to a point where adding monensin could not have a significant effect on shortening period (*Figure 4—figure supplement 1*). When the *h*-current was blocked, the period increased significantly relative to bicuculline saline (3.5 ± 0.1 s for bicuculline saline vs. 7.1 ± 0.9 s for bicuculline plus Cs$^+$ saline, n = 5, Tukey's test, p=0.003). Likewise, the interburst interval increased significantly as well relative to bicuculline saline (1.3 ± 0.05 s for bicuculline saline vs. 5.8 ± 1.0 s for bicuculline plus Cs$^+$ saline, n = 5, Tukey's test, p=0.0004). The burst duration, however, decreased significantly relative to bicuculline saline (2.2 ± 0.1 s for bicuculline saline vs. 1.3 ± 0.1 s for bicuculline plus Cs$^+$ saline, n = 5, Tukey's test, p=0.047). Adding monensin to the bicuculline plus Cs$^+$ saline did not significantly change any of the burst characteristics relative to bicuculline plus Cs$^+$ saline (*Figure 4—figure supplement 1B$_{2-4}$*). Thus, *h*-current must be present for monensin to decrease period of isolated heart interneurons (bicuculline).

Taken together, our results suggest a scenario in which the pump contributes to the normal cycling of the heart interneuron half-center oscillator by providing hyperpolarizing current that shortens the burst duration, and also the interburst interval through activation of the *h*-current. Stimulating the pump shortens the period by enhancing both of these processes. When the *h*-current is blocked, we can separate the effects of the pump on interburst interval and on burst duration. Our results are consistent with the findings of *Hill et al. (2001)* and *Sorensen et al. (2004)* whereby the *h*-current regulates the interburst interval, which in turn affects the period.

## Inhibiting the Na$^+$/K$^+$ pump with strophanthidin depolarizes the base potential, which decreases the period of half-center oscillators

In our previous experiments, we found that stimulating the Na$^+$/K$^+$ pump with monensin hyperpolarizes the membrane potential, which activates the *h*-current to shorten the period. However, *Tobin and Calabrese (2005)* found that inhibiting the pump with ouabain or with the neuropeptide myomodulin also shortened the period of these same oscillator heart interneurons just before suppressing spiking activity. We were able to reproduce their results when we performed dual extracellular recordings with a group of oscillator heart interneurons treated with strophanthidin. Within 1 min of adding 100 µM strophanthidin to the normal saline, the period decreased significantly under strophanthidin saline relative to control saline (*Figure 5A1 and A3*, 7.8 ± 0.2 s for control saline vs 5.8 ± 0.7 s for strophanthidin saline, n =5, paired t-test, p=0.0467). The duty cycle, however, increased significantly under strophanthidin saline relative to control saline (*Figure 5A1 and A4*, 57.7 ± 1.7% for control saline vs. 91.5 ± 2.8% for strophanthidin saline, n=5, paired t-test, p=0.0006). The significant increase in duty cycle was due to the significant decrease in interburst interval under strophanthidin saline relative to control saline (3.3 ± 0.1 s for control vs 0.6 ± 0.3 s for strophanthidin, n = 5, paired t-test, p=0.001). The burst duration was unchanged. We also observed the suppression of spiking activity by strophanthidin for 2–4 min before washing it out with control saline for at least 12–15 min until spiking activity resumed (*Figure 5A$_2$*). Thus, strophanthidin initially speeds up the bursting activity on oscillator heart interneurons followed by a reversible suppression of their spiking activity.

It has been well-established that inhibition of pump activity by glycosides such as ouabain or strophanthidin can further depolarize a cell (*Kerkut and Thomas, 1965*; *Carpenter and Alving, 1968*; *Smith et al., 1968*; *Baylor and Nicholls, 1969*; *Sokolove and Cooke, 1971*; *Thomas, 1972b*; *Jansen and Nicholls, 1973*; *De Weer and Geduldig, 1973*; *Mat Jais et al., 1986*; *Doebler, 2000*; *Glitsch, 2001*; *Pulver and Griffith, 2010*; *Wang et al., 2012*), which is consistent with the inward current produced by strophanthidin in our voltage-clamp experiments. Such depolarization could explain the observed decrease in the period in our experiments (*Figure 5A*) and by *Tobin and Calabrese (2005)*. To determine if the decreased period resulted from a membrane potential that has been depolarized by an inhibited pump, we performed dual intracellular and extracellular recordings to measure the base potential of oscillator heart interneurons before and after external application of 100 µM strophanthidin (*Figure 5B$_1$*). We found that strophanthidin gradually depolarized the

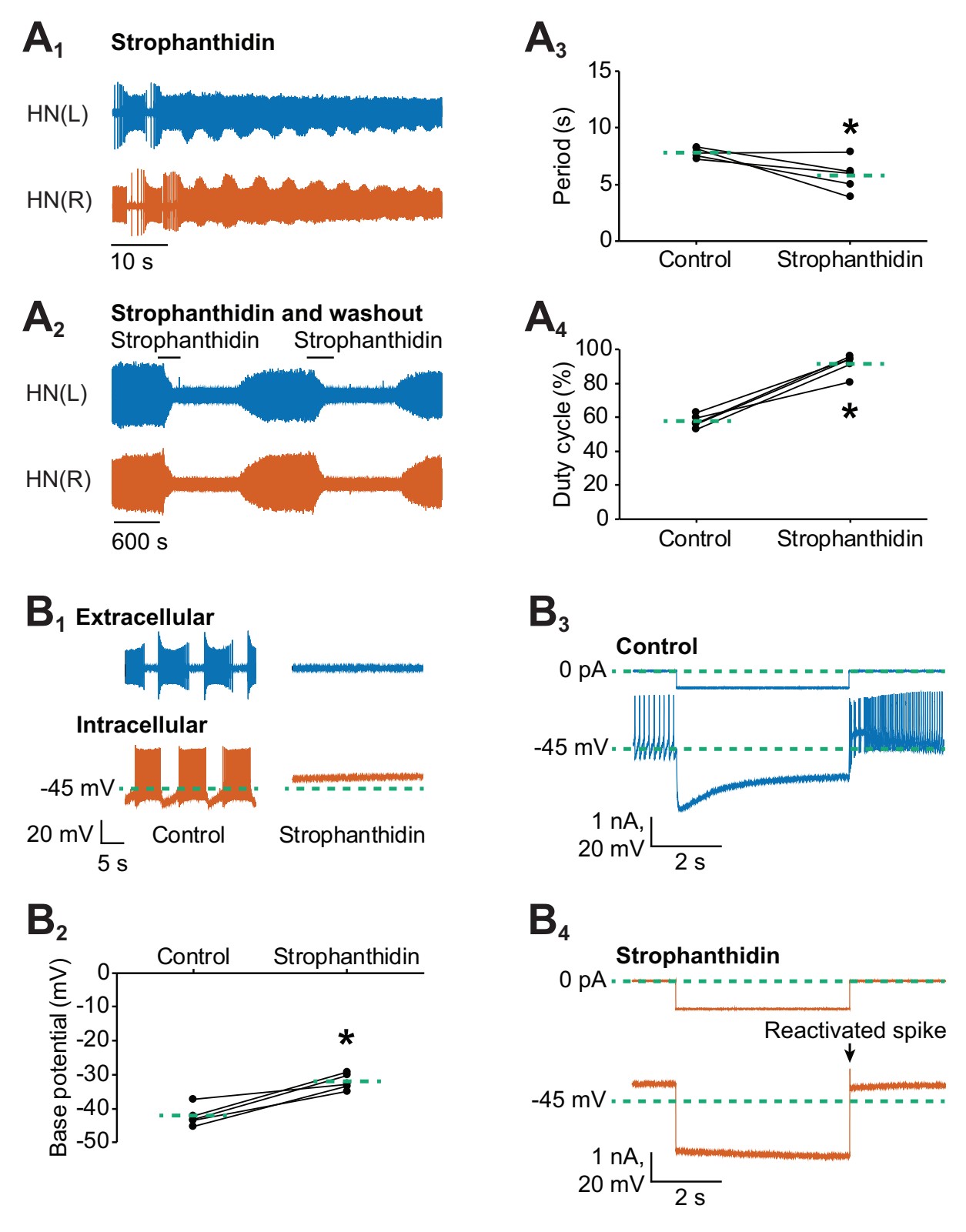

**Figure 5.** Inhibition of the Na$^+$-K$^+$ pump by applying strophanthidin speeds up the bursting activity of oscillator heart interneurons. Extracellular traces of bursting activity by left (blue) and right (vermilion) oscillator heart interneurons [HN(L) and HN(R) neurons] functioning as a half-center oscillator in (A$_1$) in saline that contained 100 μM strophanthidin. (A$_2$) Strophanthidin transiently decreased the period but eventually suppressed spiking activity, which was reversible once the neurons were again bathed in normal saline. Corresponding scatterplots of the (A$_3$) period and (A$_4$) duty cycle were

*Figure 5 continued on next page*

Figure 5 continued

measured from the extracellular traces of five preparations under ($A_{1-2}$) two experimental treatments. ($B_1$) Extracellular and intracellular traces of activity in control saline and strophanthidin saline. ($B_2$) A corresponding scatter plot of the base potential in control saline and strophanthidin saline. ($B_3$) Intracellular traces of current and voltage from an oscillator heart interneuron in normal saline. The neuron was injected with 0.6 nA for 4 s, which hyperpolarized its membrane potential below −60 mV. ($B_4$) In strophanthidin saline, the same neuron was injected with −1 nA for 4 s to hyperpolarize its membrane potential below −60 mV. The dashed green lines in the scatter plots represent means whereas the asterisks (*) represent significance from control (paired t-test, p<0.05 for all tests).

base potential of oscillator heart interneurons (data not shown), which reached an asymptote after 7 min. Compared to normal saline, the base potential under strophanthidin, which was averaged across 7–8 min, was significantly more depolarized (*Figure 5B₂*, −42.3 ± 1.4 mV for control saline vs −32.2 ± 1.1 mV for strophanthidin saline, n =5, paired t-test, p=0.003). Subtracting the base potential under strophanthidin saline from the base potential under control saline, the resting pump contributed 10.1 ± 1.6 mV to the membrane potential of oscillator heart interneurons. Thus, inhibiting the pump with strophanthidin does indeed depolarize the membrane potential of oscillator heart interneurons, which decreases the period.

To determine if the suppression of spiking activity was due to a depolarization block, we sought to reactivate spiking by injecting a short pulse (4 s) of negative current to hyperpolarize their membrane potential. Under normal saline, a negative current of −0.6 to −0.8 nA was sufficient to hyperpolarize the cells below −60 mV (*Figure 5B₃*). In strophanthidin saline, however, more negative current (−1.0 to −2.5 nA) was needed to hyperpolarize the cells beyond −60 mV (*Figure 5B₄*). Hence, the input resistance was significantly lower in strophanthidin saline relative to control saline (114.1 ± 4.5 MΩ for control saline vs. 27.7 ± 6.5 MΩ for strophanthidin saline, n=5, paired t-test, p=0.0002). These short hyperpolarizing pulses were sufficient to elicit one or more spikes in all five preparations, leading us to conclude that the suppression of spiking activity in strophanthidin saline was due to a depolarization block, consistent with previous studies (e.g., *Carpenter and Alving, 1968*; *Johnson et al., 1992*). Taken together, the decreased period that resulted from inhibiting the pump with strophanthidin was mediated by depolarization, a different mechanism from the one initiated by monensin, which activated the *h*-current.

## Inhibiting the Na⁺/K⁺ pump with K⁺-free saline decreases the period of half-center oscillators

Another well-established method to inhibit the $Na^+/K^+$ pump is to remove external $K^+$ from saline (*Garrahan and Glynn, 1967*; *Carpenter and Alving 1968*; *Maetz, 1969*; *Sokolove and Cooke, 1971*; *Thomas, 1972a*; *Gadsby 1980*; *Bühler et al., 1991*; *Lafaire and Schwarz 1986*; *Catarsi and Brunelli, 1991*; *Wang et al., 2012*). Because the pump couples the outward exchange of $Na^+$ with the inward exchange of $K^+$, the removal of external $K^+$ inhibits pump activity. To determine if the effects of strophanthidin on the bursting activity of half-center oscillator could be similarly reproduced with the removal of external $K^+$, we treated another group of oscillator heart interneurons with $K^+$-free saline. Starting with the normal bursting activity of in control (4 mM $K^+$) saline (*Figure 6A₁*), the period decreased significantly under $K^+$-free saline relative to control saline (*Figure 6A₁₋₂ and B₁₋₂*, 9.8 ± 0.9 s for control saline vs. 6.1 ± 0.2 s for $K^+$-free saline, n = 5, paired t-test, p=0.01). This effect was observed within 3–7 min of applying the $K^+$-free saline (*Figure 6B₁*). Similarly, the burst duration (*Figure 6A₁₋₂*, 5.4 ± 0.5 s for control saline vs. 3.5 ± 0.1 s for $K^+$-free saline, n=5, paired t-test, p=0.01) and interburst interval (*Figure 6A₁₋₂*, 4.4 ± 0.4 s for control saline vs. 2.6 ± 0.1 s for $K^+$-free saline, n =5, paired t-test, p=0.01) decreased significantly under $K^+$-free saline relative to control saline, resulting in no significant differences in duty cycle in both treatments. The intraburst spike frequency increased significantly under $K^+$-free saline relative to control saline (9.7 ± 1.0 Hz for control saline vs. 13.6 ± 1.1 Hz for $K^+$-free saline, n=5, paired t-test, p=0.01). Like strophanthidin, long-term exposure (16–31 min) to $K^+$-free saline also suppressed spiking activity (data not shown), which was reversible as both heart interneurons resumed their usual bursting activity after 1–5 min of washing out the $K^+$-free saline with normal saline (data not shown). Thus, the suppressive effects of $K^+$-free saline are also reversible.

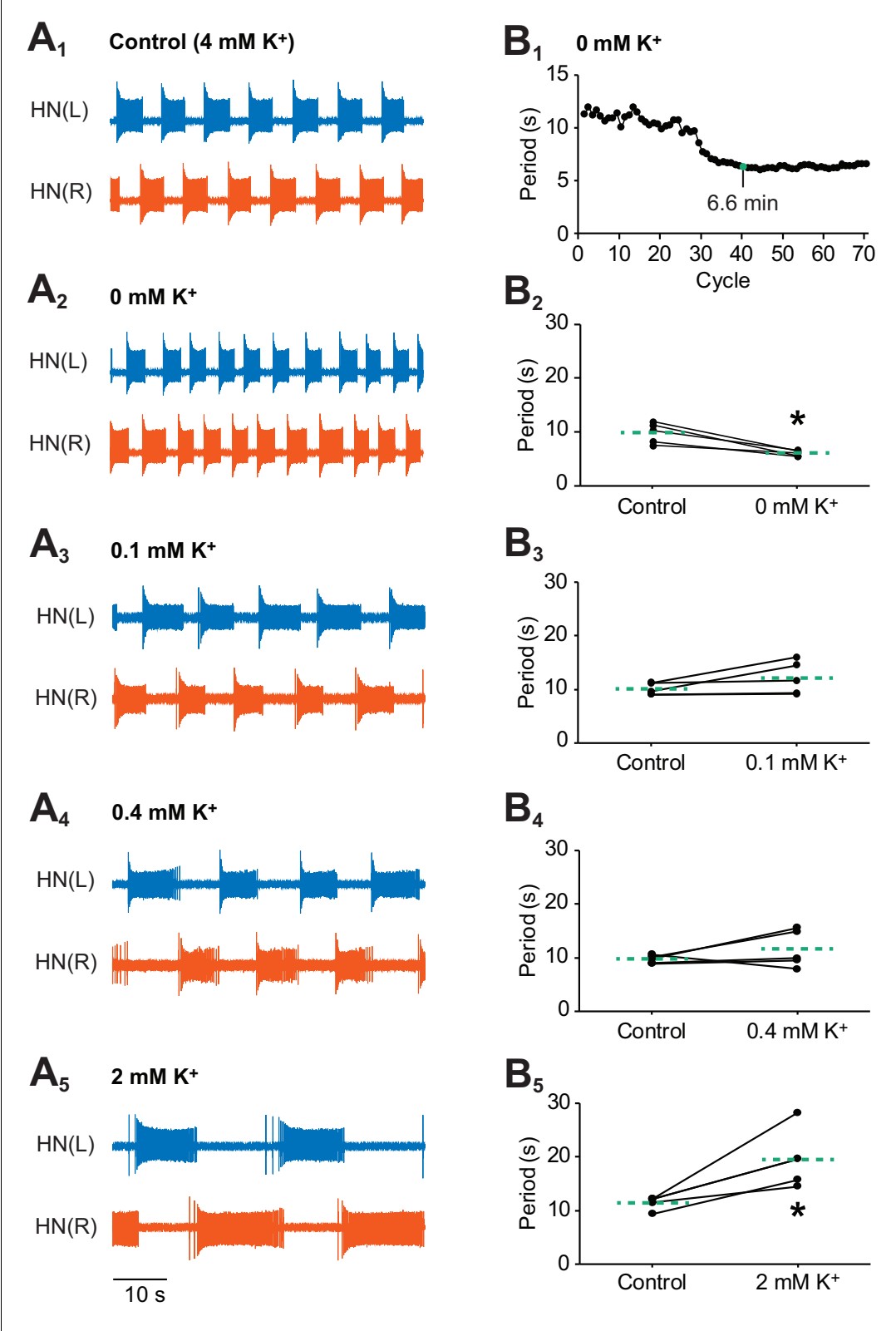

**Figure 6.** Effects of various concentrations of external K$^+$ on the bursting activity of oscillator heart interneurons. Extracellular traces of bursting activity by left (blue) and right (vermilion) oscillator heart interneurons [HN(L) and HN(R) neurons] treated with ($A_1$) control (4 mM K$^+$) saline, ($A_2$) 0 mM, ($A_3$) 0.1 mM, ($A_4$) 0.4 mM, and ($A_4$) 2 mM. ($B_1$) A period vs cycle graph of one preparation showing the shortening of the period in K$^+$-free saline over time. Summing up all the periods leading up to the 40th cycle reveals that it takes 6.6 min for the effects of K$^+$-free saline to stabilize. Scatterplots of

*Figure 6 continued on next page*

*Figure 6 continued*

corresponding periods for (B$_2$) 0 mM, (B$_3$) 0.1 mM, (B$_3$) 0.4 mM, and (B$_5$) 2mM K$^+$ were measured from the extracellular traces of the four groups treated with (A$_{2-5}$) lower concentrations of external K$^+$. The dashed green lines represent means whereas the asterisks (*) represent significance from control (paired t-test, p<0.05 for all tests).

To exclude the possibility that the decreased period under K$^+$-free saline was mainly due to a change in the equilibrium potential for K$^+$, we performed additional extracellular experiments with varying low concentrations (0.1, 0.4, and 2 mM) of external K$^+$ (*Figure 6A$_{3-5}$*). Unlike the period in K$^+$-free saline, the period did not decrease significantly when the oscillator heart interneurons were exposed to any of the three low concentrations of external K$^+$ (*Figure 6B$_{3-5}$*). In fact, the period either remained the same (0.1 and 0.4 mM K$^+$ saline, *Figure 6A$_{3-4}$ and B$_{3-4}$*) or increased significantly (2 mM K$^+$ saline, *Figure 6A5 and B5*, 11.4 ± 0.5 s for control saline vs. 19.6 ± 2.4 s for K$^+$-free saline, n=5, Tukey's test, p=0.005). Only in K$^+$-free saline, did we observe a significantly decreased period. Thus, it is the inhibition of the pump and not the change in the equilibrium potential for K$^+$ that is primarily responsible for the initial increase in bursting activity.

## An extended model of heart interneuron half-center oscillator incorporating the Na$^+$/K$^+$ pump

To model the bursting activity of the heart interneuron half-center oscillator and to explain mechanistically the results of our monensin experiments, we modified the biophysical model by *Hill et al. (2001)* by introducing the Na$^+$/K$^+$ pump current and intracellular Na$^+$ dynamics (See *Equations 4 and 5* in Materials and methods). The monensin-mediated diffusion of Na$^+$ across the cell membrane was modeled as a non-electrogenic process controlled by a monensin rate constant (*M*), which contributed to intracellular Na$^+$ dynamics. To determine the extent to which the bursting activity of our model captures the bursting activity observed in our experiments, we compared the modeled data to the experimental data based on four burst characteristics (period, burst duration, interburst interval, and duty cycle) that mimicked our three experimental treatments (*Figure 4A$_{1-3}$*, *Table 1*). The first experimental treatment was a half-center oscillator (control saline, *Figures 4A1* and *7A1*), which was modeled by including expressions for all voltage-gated and synaptic currents. The second experimental treatment was a half-center oscillator with the pump stimulated by monensin (monensin saline, *Figures 4A2* and *7A2*), which was simulated by setting the monensin rate constant to 2.2 × 10$^{-3}$s$^{-1}$ (these values have been rounded; see the Appendix for the exact values used in our simulations). Finally, the third experimental treatment was a half-center oscillator that has a pump stimulated by monensin and a blocked *h*-current (monensin plus Cs$^+$ saline, *Figures 4A3* and *7A2*), which was simulated by reducing the maximal conductance of the *h*-current ($\bar{g}_h$ = 0.1 nS) and by setting the monensin rate constant to 1.9 × 10$^{-4}$ s$^{-1}$.

The monensin rate constants in the above two modeled experimental treatments were chosen to provide the best fit to our experimental data. Both rate constants can be found close to the boundaries of their respective models (e.g., 2.2 × 10$^{-3}$ s$^{-1}$ for a half-center oscillator in *Figure 7B$_1$*) and differed from each other by an order of magnitude (10$^{-3}$ s$^{-1}$ vs. 10$^{-3}$ s$^{-1}$) because the models themselves had different sensitivities to the monensin rate constant, with the half-center oscillator with a blocked *h*-current being more sensitive. Increasing the monensin rate constant towards its boundary value generally decreases the period, and spiking activity was suppressed when the rate constants were at values greater than its boundary value.

To determine if our simulations captured the trends observed in our experiments, we compared the burst characteristics of our model to the ones from our experiments. We used the control data of all extracellular experiments involving monensin. When we simulated the experimental treatments in our model, all the burst characteristics were within range except for the period, duty cycle, and interburst interval from our model of a half-center oscillator with a stimulated pump (*Table 1*). Thus, nine of the twelve measures (four burst characteristics multiplied by three experimental treatments) were within range of the experimentally observed characteristics. In cases where the burst characteristics were out of range, the simulations still qualitatively captured the trends observed in our experiments whereby stimulation of the pump by monensin speeds up the bursting activity of oscillator heart interneurons when the *h*-current is present.

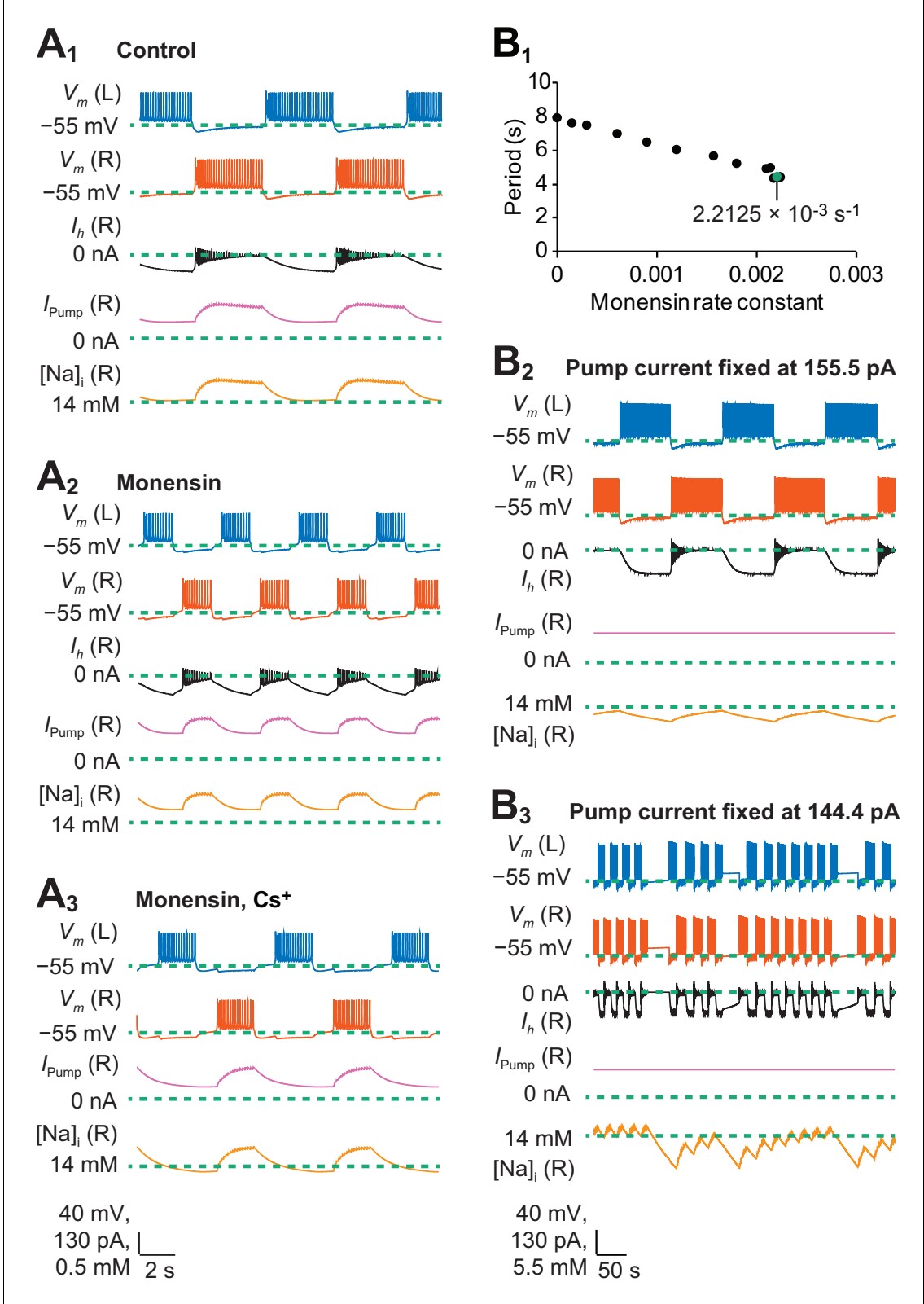

**Figure 7.** A biophysical model of oscillator heart interneurons that mimics three experimental treatments with monensin. (A₁) Sample traces of simulated activity by oscillator heart interneurons functioning as a half center oscillator in normal saline, which were observed when parameters $\bar{g}_h$ = 4.9 nS and $M$ = 0 s⁻¹. (A₂) Simulated activity of oscillator heart interneurons with a Na⁺/K⁺ pump stimulated by monensin (monensin saline), observed when $\bar{g}_h$ = 4.9 nS and $M$ = 2.2 × 10⁻³ s⁻¹. (A₃) Simulated activity of a half-center oscillator with blocked $h$-current and pump stimulated by monensin

*Figure 7 continued on next page*

*Figure 7 continued*

(monensin plus $Cs^+$ saline), observed when $\bar{g}_h$ = 0.1 nS and $M$ = 1.9 × $10^{-4}$ $s^{-1}$. Sample traces representing membrane potentials ($V_m$) of both left (L, blue) and right (R, vermilion) oscillator heart interneurons as well as $h$-current ($I_h$, yellow), pump current ($I_{pump}$, reddish purple), and intracellular $Na^+$ concentration $[Na]_i$ (black) belonging to the right oscillator heart interneuron. ($B_1$) A scatterplot depicting incremental shortening of the period as the monensin rate constant increases towards 2.2 × $10^{-3}$ $s^{-1}$ in a model of a half-center oscillator with the $h$-current present. Spiking activity was suppressed at rate constant values larger than 2.2 × $10^{-3}$ $s^{-1}$. ($B_2$) In a simulation of a half-center oscillator, the average pump current over the entire burst cycle was fixed at 155.5 pA, which resulted in a longer period, burst duration, and interburst interval. Intracellular $Na^+$ concentration $[Na]_i$ appeared to increase and decrease more slowly relative to a ($A_1$) normal half-center oscillator with a dynamic pump current. ($B_3$) Fixing the average pump current over the entire burst cycle to 144.4 pA (a value lower than 155.5 pA) produced irregular bouts of bursting.

## The heart interneuron half-center oscillator model explains how the interaction between the $Na^+/K^+$ pump current and the $h$-current influences the period and interburst interval

In our experiments, stimulating the $Na^+/K^+$ pump with monensin will only decrease the period of half-center oscillators if the $h$-current was present (*Figure 4A$_{1-2}$ and B$_1$*), revealing an interaction between the pump current and the $h$-current to regulate bursting. To explain this interaction further using our model, we analyzed changes in the pump current and the $h$-current during each burst and interburst interval. We computed average values for the pump current over a burst and over an interburst interval. We only computed the average value for the $h$-current over an interburst interval because the $h$-current activates only when hyperpolarized but deactivates when depolarized. Average values for both currents were obtained by computing the integral of each current over a closed interval and then dividing that integral by the duration of the closed interval. When averaging these currents, we only considered the segment of the interburst interval from the time after the last action potential in a burst in which the membrane potential crossed −50 mV to the time before the first action potential of the next burst whereby the membrane potential crossed −50 mV again. This segment was chosen to avoid having substantial parts of each current originating from the end of a plateau of a previous burst and the beginning of a plateau of the next burst.

When we set the monensin rate constant to 2.2 × $10^{-3}$ $s^{-1}$ to simulate the monensin treatment in our half-center oscillator model, each burst duration decreased from 3.7 s to 1.6 s (*Table 1*, *Figure 7A$_{1-2}$*), which we attributed to the average pump current during each burst, which increased from 183.6 pA to 221.6 pA. The increased pump current itself can be attributed to the average increase in intracellular $Na^+$ concentration during each burst from 14.4 to 14.6 mM. Similarly, adding monensin also decreased each interburst interval from 4.2 s to 2.8 s (*Table 1*, *Figure 7A$_{1-2}$*) but increased the average intracellular concentration during each interburst interval from 14.1 to 14.4 mM, which in turn increased the average pump current during each interburst interval from 108.3 pA to 166.9 pA. As a result, the average membrane potential during each interburst interval hyperpolarized from −60.6 mV to −62.2 mV, which increased the average magnitude of the $h$-current during each interburst interval from −69.9 pA to −82.6 pA. When we reduced the monensin

**Table 1.** Modeled burst characteristics (period, burst duration, interburst interval, and duty cycle) were compared to experimental ranges under three parameter regimes that mimicked our experimental treatments. The control treatment was a normal half-center oscillator. The monensin treatment was a half-center oscillator with a pump stimulated by monensin ($M$ = 2.2 × $10^{-3}$ $s^{-1}$). The monensin plus $Cs^+$ treatment was a half-center oscillator with blocked $h$-current and a pump stimulated by monensin ($M$ = 1.9 × $10^{-4}$ $s^{-1}$). Asterisks denote out-of-range values.

**Comparison of modelled burst characteristics to experimental ranges under four parameter regimes.**

| Treatments | Period (s) | | Burst duration (s) | | Interburst interval (s) | | Duty cycle (%) | |
|---|---|---|---|---|---|---|---|---|
| | Model | Experimental ranges | Model | Experimental ranges | Model | Experimental ranges | Model | Experimental ranges |
| Control | 8.0 | 4.3-12.3 | 3.7 | 2.6-7.3 | 4.2 | 1.5-6.0 | 47.0 | 46.8-64.7 |
| Monensin | 4.4* | 2.5-4.1 | 1.6 | 1.3-2.3 | 2.8* | 1.1-1.8 | 35.5* | 44.6-59.8 |
| Monensin, $Cs^+$ | 6.6 | 5.9-8.6 | 2.0 | 1.7-4.1 | 4.5 | 3.3-5.5 | 30.9 | 26.6-48.0 |

rate constant to $1.9 \times 10^{-4}$ $s^{-1}$ and the maximal conductance of the $h$-current to 0.1 nS, each burst duration increased from 1.6 s to 2.0 s and each interburst interval increased from 2.8 s to 4.5 s (**Table 1**, **Figure 7A$_{2-3}$**). Blocking the $h$-current in our monensin simulation also decreased the average pump current during each burst from 221.6 pA to 153.0 pA and during each interburst interval from 166.9 pA to 92.1 pA. The decreased pump current was due to the average decrease in intracellular $Na^+$ concentration from 14.6 mM to 14.3 mM during each burst and from 14.4 mM to 14.0 mM during each interburst interval. Taken together, the application of monensin in our simulations decreased the burst duration by increasing the average pump current during each burst. Moreover, monensin also increased the average pump current during each interburst interval, which hyperpolarized the membrane potential and activated the $h$-current during each interburst interval, which in turn decreases the interburst interval itself. Our simulations also confirmed our experimental results whereby stimulating the pump while blocking the $h$-current in half-center oscillators shortens the burst duration but increases the interburst interval, which has the effect of decreasing the duty cycle. These simulations confirmed our experimental results that a stimulated pump interacts with the $h$-current to decrease the period that results from the combined decrease of the burst duration and interburst interval.

## A dynamic $Na^+/K^+$ pump current is essential for normal bursting activity in simulations of a half-center oscillator

To determine the dynamic contributions of the $Na^+/K^+$ pump current and intracellular $Na^+$ concentrations to the bursting activity of our modeled half-center oscillator, we examined the changes of the pump current and intracellular $Na^+$ concentration during specific time points of each burst and interburst interval. The pump current at the end of each interburst interval had a minimum (or baseline) value of about 98.0 pA (**Figure 7A$_1$**). During the beginning of each burst, intracellular $Na^+$ concentration accumulated with each action potential (**Figure 7A$_1$**). As the intracellular $Na^+$ concentration increased, the pump current also increased. Over the course of a burst, the spike frequency decreased and the outward $Na^+$ flux generated by the pump began to balance—at a maximum pump current of 203.6 pA —and exceeded the inward $Na^+$ flux produced by the voltage-gated $Na^+$ currents (**Figure 7A$_1$**). At the termination of each burst, the intracellular $Na^+$ concentration and pump current relaxed to a baseline of about 14 mM and 100 pA, respectively, over the duration of an interburst interval (**Figure 7A$_1$**). The outward pump current was active over the entire cycle of bursting activity, but it was stronger during each burst. These effects are much like the ones we would expect from a leak current with a very negative equilibrium potential: an outward current that seems to track the membrane potential exerting a hyperpolarizing drag on both the burst duration and the interburst interval. The pump current actually tracks intracellular $Na^+$ dynamics, which is delayed (low-pass filtered) with respect to the membrane potential and this delay might exert sizable effects in some cases.

To determine if a dynamic pump current is essential to the normal bursting activity of oscillator heart interneurons, we performed simulations whereby the average pump current was fixed at a constant value of 155.5 pA over the entire burst cycle (**Figure 7B$_2$**). Compared to the dynamic pump current, the value of this constant pump current was lower during each burst but higher during each interburst interval. This constant value was chosen because lower values of the pump current (e.g., 144.4 pA) produced irregular bouts of bursting (**Figure 7B$_3$**). When the pump current was fixed to 155.5 pA, the period, burst duration and interburst interval were 19.9 s, 9.6 s, and 10.2 s, respectively (**Figure 7B$_2$**). The durations of these burst characteristics were longer than those observed in simulations with a dynamic pump current (**Figure 7A$_1$** and **Table 1**). The relatively lower constant pump current during each burst increased the burst duration. Because the period in the half center oscillator is determined in part by the burst duration, the period itself became longer relative to simulations with a dynamic pump. Taken together, we concluded that in a half-center oscillator, a dynamic pump current that changes from an interburst interval to the next burst is critical for the formation of burst patterns with normal burst characteristics. Such a conclusion is also consistent with the observed non-spiking activity that result from inhibiting the pump with strophanthidin or $K^+$-free saline in our experiments.

## Discussion

### Effects of stimulating the Na⁺/K⁺ pump in neurons that pace the leech heartbeat central pattern generator

In the present study, we examined the role of the $Na^+/K^+$ pump in regulating the bursting activity of oscillator heart interneurons, which pace the leech heartbeat central pattern generator. In doing so, we performed experiments that allowed us to determine the effects of a stimulated pump on the burst characteristics of these oscillator heart interneurons (*Figures 3–4*). Because the rate of pump activity is dependent on intracellular $Na^+$ concentrations (*Thomas, 1972a*; *Mogul et al., 1990*), we were able to stimulate the pump of the oscillator heart interneurons by increasing the intracellular loading of $Na^+$ with an electrode (*Figure 1A₂*) or by applying monensin (*Figures 1B3* and *2A2*), a $Na^+/H^+$ antiporter (*Hill and Licis, 1982*). Monensin has a hyperpolarizing effect when applied to excitable cells such as neurons and muscle cells (*Lichtshtein et al., 1979*; *Tsuchida and Otomo 1990*, *Satoh and Tsuchida, 1999*; *Doebler, 2000*; *Wang et al., 2012*). Our stimulation of the pump with monensin did indeed hyperpolarize the membrane potential of the oscillator heart interneurons when the *h*-current and $Ca^{2+}$ currents were blocked (*Figure 1B₃*). When we voltage-clamped oscillator heart interneurons under the same conditions, we observed that monensin produced a long-lasting outward current, indicating that the outward pump current was enhanced by monensin (*Figure 2A₁₋₂*). This outward current is consistent with the results of *Tsuchida and Otomo (1990)*, who also showed monensin enhancing the outward current generated by Purkinje fibers that were isolated from the hearts of dogs and goats. Moreover, the monensin-enhanced outward pump current can be blocked by strophanthidin, which is consistent with previous studies that found glycosides such as ouabain and strophanthidin blocked the hyperpolarizing effects of monensin (*Lichtshtein et al., 1979*; *Doebler, 2000*; *Wang et al., 2012*).

Since we did not see any corresponding change in the membrane conductance that was assessed from the linear slope of each current-voltage curve (*Figure 2B₂*), it is unlikely that the observed outward shift in the membrane current is due to other voltage-gated or leak currents that are intrinsic to these oscillator heart interneurons. Our voltage-clamp experiments were performed in $Ca^{2+}$-free saline that contained $Mn^{2+}$, which would have prevented the activation of $Ca^{2+}$-activated $K^+$ current. Even if the $Ca^{2+}$-activated $K^+$ current was present, it is a fast transient current in these neurons, which could not sustain a long-term outward current (*Simon et al., 1992*). The combined use of strophanthidin and monensin in our voltage-clamp experiments allowed us to determine the fixed range (202–289 pA) of the maximum pump current by taking the difference between the outward current produced by monensin and the inward current produced by monensin plus strophanthidin. At rest, the pump current varies between 82 pA to 283 pA, a range that is consistent with outward currents attributed to the pump at rest in other voltage-clamp experiments (e.g., *Sakai et al., 1996*; *Mogul et al., 1990*). Because the maximum pump current is usually not generated under resting conditions, there is a certain amount of pump current left available that can be stimulated by monensin. These results indicate that the action of monensin in oscillator heart interneurons is consistent with its well-established role of facilitating an electroneutral exchange of intracellular $H^+$ for extracellular $Na^+$, thereby increasing intracellular $Na^+$, which in turn stimulates the pump to generate more outward current that hyperpolarizes the membrane potential (*Lichtshtein et al. 1979*).

A possible side effect of the continuous extrusion of $H^+$ by monensin is a modest change in intracellular pH (*Smallridge et al., 1992*; *Itoh et al., 2000*). For example, at the same concentration of 10 µM used in our experiments, monensin has been found to increase the intracellular pH of astroglial cells by 0.07 (from 7.26 to 7.33) (*Itoh et al, 2000*). Intracellular alkalization has been reported to prolong the bursting activity of ventral white cells in snails, increase the number of spikelets in hippocampal neurons of rodents, decrease the viability of endothelial cells, and enhance glycolysis in neurons and glioma cells (*Gillette, 1983*; *Valiante et al., 1995*; *Cutaia et al., 2000*). In our experiments, we did not observe any prolongation of bursting activity (*Figure 4B₁*) or an increase in intraburst spike frequency (*Figure 4B₅*). Moreover, the monensin-treated oscillator heart interneurons do not appear to be adversely affected by any potential intracellular alkalization as they were able to burst continuously for well over thirty minutes to an hour (data not shown). Nevertheless, if monensin were to enhance glycolytic functions, thereby increasing ATP production, it would only serve to enhance pump activity (*Lombardo et al., 2004*), which would be consistent with the use of monensin

in this study. Thus, even if modest intracellular pH changes by monensin occurred, we do not expect these changes to confound our results or to affect our interpretations of them.

## Na$^+$/K$^+$ pump interacts with the *h*-current to speed up bursting in heartbeat central pattern generator neurons

A notable effect of stimulating the pump of oscillator heart interneurons with monensin was the significant decrease in their period (*Figures 4A$_1$ and B$_1$*). This acceleration was unexpected because a stimulated or reactivated Na$^+$/K$^+$ pump has been consistently demonstrated in oscillator heart interneurons (*Figures 1–2*) and in other tissues and systems (e.g., *De Weer and Geduldig, 1973*; *Eisner and Lederer, 1980*; *Johnson et al., 1992*) to cause hyperpolarization. The paradox of stimulating the pump to speed up bursting activity can be explained if we consider other intrinsic ionic currents that are also present and are likely to be activated or deinactivated by a membrane potential hyperpolarized by an outward pump current. The *h*-current is an obvious candidate as it is an inward current with a voltage-threshold near -50 mV and becomes fully activated at −70 to −80 mV (*Angstadt and Calabrese, 1989*). The threshold of the *h*-current is well within the range of membrane potential produced by monensin. The role of the *h*-current in controlling the rate of depolarization after hyperpolarization has been demonstrated experimentally in neurons and cardiac pacemakers that oscillate continuously (*Soltesz et al., 1991*; *DiFrancesco 1993*; *Bal and McCormick, 1997*; *Zhang et al., 2003*). Moreover, based on the biophysical model of oscillator heart interneurons by *Hill et al. (2001)*, increasing the *h*-current from its canonical value produces a substantial decrease in period. Their findings were later confirmed by *Sorensen et al. (2004)*, who found that increasing the *h*-current conductance of living or silicon neurons with dynamic clamp decreases the period and interburst interval, without affecting the burst duration. Thus, activation of the *h*-current can explain the non-intuitive effects of monensin in shortening the period of central pattern generator neurons.

The possible interaction between the pump current and the *h*-current has been suggested by previous studies on slow afterhyperpolarizations (e.g., *Robert and Jirounek, 1998*; *Soleng et al., 2003*; *Baginskas et al., 2009*). For example, when *Robert and Jirounek (1998)* recorded the slow afterhyperpolarization (described by the authors as a slow posttetanic hyperpolarization), which occurs after a burst of compound action potentials, in the rabbit vagus nerve, they found that blocking the *h*-current or the inward rectifier K$^+$ with Cs$^+$ and Ba$^{2+}$, respectively, would increase the amplitude of the slow afterhyperpolarization. Because the slow afterhyperpolarization has been documented to be a product of the pump current, *Robert and Jirounek (1998)* hypothesized that the *h*-current and the inward rectifier K$^+$ current may serve to counterbalance the effects of the pump current under normal conditions. Such a counterbalancing role by the *h*-current could explain the previously mentioned paradoxical shortening of the period when we stimulated the pump in the heart interneurons. We were able to confirm this role of the *h*-current in the shortening of the period by monensin when we blocked the *h*-current with Cs$^+$, which prevented the period from shortening (*Figure 4B$_1$*). Similar results were also observed when the oscillator heart interneurons functioned as individual bursters when pharmacologically isolated with bicuculline (*Figure 4—figure supplement 1*). Thus, our experimental results support our hypothesis that stimulation of the Na$^+$/K$^+$ pump in oscillator heart interneurons activates the *h*-current, which in turns speeds up rhythmic bursting in half-center oscillators and isolated interneurons.

When we blocked the *h*-current with Cs$^+$ while stimulating the pump with monensin in our extracellular experiments, we found that the interburst interval increased significantly, consistent with the established role of the *h*-current in regulating the interburst interval of oscillator heart interneurons (*Hill et al., 2001*; *Sorensen et al., 2004*). The burst duration, however, remained the same when the *h*-current was blocked. Thus, the *h*-current regulates the period by changing the interburst interval, with a shorter period resulting from a decreased interburst interval. Because the burst duration and interburst interval decreased proportionally in monensin saline relative to control saline (*Figure 4B$_{2-3}$*), the duty cycle in monensin saline remained similar to the one under control saline (*Figure 4B$_4$*). It was only when the *h*-current was blocked with Cs$^+$ that we observed a decrease in the duty cycle that resulted an increased interburst interval (*Figure 4B$_3$*). Conversely, *Zhang et al. (2003)* found that overexpression of the *h*-current in pyloric dilator neurons of the stomatogastric ganglion increased the duty cycle of these neurons. Thus, the pump current regulates the burst duration

whereas the *h*-current regulates the interburst interval and changes to one or both of these two burst characteristics affect the period and duty cycle of oscillator heart interneurons.

In summary, we posit a sequence of events by which stimulation of the pump current with monensin activates the *h*-current to speed up the bursting activity in a half-center oscillator. First, monensin increases the intracellular loading of $Na^+$ as a result of its electroneutral exchange of $H^+$ for $Na^+$ across the cell membrane. Second, the increased intracellular loading of $Na^+$ stimulates the pump to generate more outward current. Third, the enhanced outward current hyperpolarizes the membrane potential of the neuron, which activates the *h*-current. Finally, activation of the *h*-current depolarizes the membrane potential, which promotes faster recovery from inhibition and speeds up the bursting activity of half-center oscillators.

### Inhibiting the $Na^+/K^+$ pump depolarizes the membrane potential to speed up the bursting activity of oscillator heart interneurons

One seeming paradox of manipulating the $Na^+/K^+$ pump in oscillator heart interneurons current is the increase in bursting activity, regardless of whether the pump has been stimulated or inhibited. *Tobin and Calabrese (2005)* first observed that inhibiting the pump with the neuropeptide myomodulin or with the glycoside ouabain shortened the period of oscillator heart interneurons. In contrast to monensin, myomodulin produced an inward shift in the ramp current of oscillator heart interneurons when the *h*-current and $Ca^{2+}$ currents were blocked (*Tobin and Calabrese, 2005*). When ouabain was present, myomodulin could not produce further inwards shifts in the ramp current, indicating that myomodulin inhibits the pump like ouabain. The results from our strophanthidin experiments are consistent with their observations as we also observed a decrease in the period followed by the suppression of spiking activity in oscillator heart interneurons treated with strophanthidin. Furthermore, we confirmed that the suppression of spiking activity was due to a depolarization block, consistent with previous studies (e.g., *Rhoades and Gross, 1994*; *Krey et al., 2010*). Unlike ouabain, the effects of strophanthidin are reversible in leech neurons (*Baylor and Nicholls, 1969*), and spiking resumed once the neurons were washed out with normal saline.

We were able to reproduce the results of our strophanthidin experiments using $K^+$-free saline, whereby the period initially decreases followed by the suppression of spiking. In contrast, the periods of neurons bathed in lower concentrations of external $K^+$ either increased or remained the same. Thus, it appears that the decreased period in $K^+$-free saline was due to the depolarized membrane potential that resulted from an inhibited pump, whereas the increased or unchanged period observed in low $K^+$ (0.1–2 mM) saline was due to a more negative equilibrium potential for $K^+$. Such a conclusion is consistent with the findings of *Carpenter and Alving (1968)*, who found that the resting membrane potential of $R_2$ neurons in *Aplysia californica* was more depolarized when bathed in $K^+$-free seawater at 20°C. But when they lowered the external concentration of $K^+$ to just 1 mM, the $R_2$ neurons hyperpolarized instead.

In summary, the paradox of increased bursting activity following the inhibition or stimulation of the $Na^+/K^+$ pump in oscillator heart interneurons can be explained by two different mechanisms initiated by pump stimulation and by pump inhibition. Pump stimulation hyperpolarizes the membrane potential, which activates the *h*-current to shorten the period whereas pump inhibition depolarizes the membrane potential, increasing the likelihood of spiking activity, thereby also shortening the period. Thus, the activation of these two separate pathways ultimately result in the same decrease in period. It is also interesting to note that myomodulin not only inhibits the pump current but increases the *h*-current thus ensuring that there is ample *h*-current during each burst cycle despite the depolarization caused by reduction of the pump current (*Tobin and Calabrese, 2005*).

### A biophysical model that includes the $Na^+/K^+$ pump current and intracellular $Na^+$ dynamics captures our experimental findings

Our biophysical model, which includes the $Na^+/K^+$ pump and intracellular $Na^+$ dynamics, captured and explained the changes in each of the four burst characteristics (period, burst duration, interburst interval, and duty cycle) within each of the three experimental treatments. Our model also confirmed our experimental results showing that increasing intracellular $Na^+$ concentrations enhances the hyperpolarizing pump current, which interacts with the *h*-current to shorten the interburst interval (*Figures 7A$_2$*). Moreover, the model suggests that when a half-center oscillator model is treated

with monensin, the pump current contributes to burst termination thereby influencing burst duration (*Figure 7A₂*). Overall, our model quantitatively reproduced nine of the twelve measures (four burst characteristics multiplied by three experimental treatments). Although the three modeled burst characteristics did not fall within the experimentally observed range, the model still faithfully reproduced the observed trends of the experimental data. Our model also shows that a dynamic pump current is essential to normal bursting activity. When we set the average pump current to be constant over the entire burst cycle, the period of the oscillator heart interneuron simulations became longer, as each burst duration and interburst interval increases. Thus, our model reveals that importance of a pump current that is dynamic and not static for there to be normal bursting activity in the oscillator heart interneurons. Furthermore, stimulating the pump current with monensin offsets the amount pump current being generated, thereby influencing the entire burst cycle.

The ability of our model to account for multiple experimental treatments is notable because models that do so are generally expected to better predict dynamics under a wide range of other treatment conditions. These other treatment conditions may include modulation or removal of subsets of currents that elicit characteristic oscillatory behaviors such as seizure-like activities or slow subthreshold oscillations (*Grillner et al., 1995*; *Cymbalyuk and Calabrese, 2000*, *2001*; *Krishnan and Bazhenov, 2011*; *Jasinski et al., 2013*; *Barnett et al., 2013*; *Bicanski et al., 2013*; *Doloc-Mihu and Calabrese, 2014*; *Krishnan et al., 2015*). Moreover, the greater sensitivity of the model without $h$-current to the monensin rate constant suggests that the $h$-current plays a protective role to prevent disruption of functional activity in oscillator heart interneurons, consistent with previous observations (*Cymbalyuk et al., 2002*; *Marin et al., 2013*).

## The Na$^+$/K$^+$ pump affects bursting activity in a similar fashion to a leak current

The action of the Na$^+$/K$^+$ pump current in bursting neurons and networks is very similar to the action of a leak current. The leak current is predominantly an outward current and if its reversal potential is sufficiently hyperpolarized, it is an outward current over the entire burst cycle. *Cymbalyuk et al. (2002)* found that the conductance and reversal potential of the leak current controlled properties of bursting activity in isolated oscillator heart interneurons. For example, when the leak conductance increases, the duty cycle and burst duration decrease (*Cymbalyuk et al., 2002*). The pump current is always an outward current that increases, along with the leak current, during each burst. In simulations of a half-center oscillator, over the course of the burst this outward current creates a drag that promotes burst termination as the underlying inward currents wane. An increase in an outward current, such as a leak current or a pump current, or increase of a constant outward current, could lead to earlier burst termination. In contrast, the interburst interval of a half-center oscillator is mainly determined by the interaction of the $h$-current with the pump current but also by the burst duration of the inhibiting cell. Adding monensin in our model slightly increases the intracellular Na$^+$ concentration throughout the burst cycle. By increasing the baseline intracellular Na$^+$ concentration and the resulting pump current in both phases of each burst cycle, we achieved an effect similar to introducing an outward leak current.

The time course of activation of the pump current differentiates it from a leak current. Because the leak current is an instantaneous function of membrane voltage, its waveform tracks the membrane potential exactly. The pump current activates incrementally as action potentials contribute to the intracellular Na$^+$ concentration. As the pump current gradually builds, it can aid in the termination of each burst. After each burst, the intracellular Na$^+$ concentration takes time to decay to its baseline between bursts, resulting in a strong outward pump current being present immediately after each burst, which hyperpolarizes the membrane potential and activates the $h$-current during each interburst interval.

## Contributions of the Na$^+$/K$^+$ pump current to motor activity

The Na$^+$/K$^+$ pump is principally credited with maintaining the intracellular concentrations of Na$^+$ and K$^+$ (*Ritchie, 1971*; *Clausen, 2005*). Although the pump generates an outward current, its contribution to the resting membrane potential is often too small to be detectable (*Thomas, 1972a*). Only recently has the potential role of the pump current in regulating motor activity been explored, and there is growing experimental evidence that the pump current can contribute to the motor activity

(e.g., *Ballerini et al., 1997*; *Tobin and Calabrese, 2005*; *Krey et al., 2010*; *Pulver and Griffith, 2010*; *Zhang and Sillar, 2012*; *Zhang et al., 2015*). In *Drosophila* larval motor neurons (*Pulver and Griffith, 2010*) and in spinal central pattern generator neurons of *Xenopus* tadpoles (*Zhang and Sillar, 2012*), the pump current generates slow afterhyperpolarizations, which are long-lasting and are thought to be involved in short-term motor memory through the integration of spikes. The contribution of the pump current to motor activity may also manifest itself through complex dynamical interactions with other ionic currents (*Pulver and Griffith, 2010*; *Zhang et al., 2015*). *Pulver and Griffith (2010)* observed that the pump-mediated slow hyperpolarization in *Drosophila* larval motor neurons releases the A-current from inactivation, thereby modifying the neuron's response to the next depolarizing input by delaying the initiation of the spike. This delay could be abolished by ouabain or K$^+$-free saline, as shown by *Zhang et al. (2015)* using *Xenopus*.

Unlike the A-current in *Drosophila* or *Xenopus*, the A-current in leech heart interneurons has short activation and inactivation kinetics, in the range of milliseconds, and is therefore unlikely to sustain an effect on the period, which lasts for seconds. Instead, our results point to an interaction between the pump current and the *h*-current to regulate ongoing bursting activity of central pattern generator neurons. Such regulation has strong behavioral implications as changes in burst characteristics underlie the production of various rhythmic behaviors (e.g., walking vs. running or breathing fast vs. breathing slow) (*Hooper, 1998*). Nevertheless, for the pump current to regulate bursting activity, it has to be dynamic and stimulating the pump currents offsets the amount of pump current generated. Thus, the dynamics of the pump current, along with the dynamics of the *h*-current, can help shape the bursting activity of neurons and networks that control motor output.

## Materials and methods

### Preparation

Medicinal leeches, *Hirudo* spp. Linneaus 1758, weighing between 1–1.5 g each, were obtained from Niagara Leeches (Cheyenne, WY) and Leeches USA (Westbury, NY) and were maintained in artificial pond water [0.05% (w/v) Instant Ocean sea salt (Spectrum Brands Inc., Madison, WI) diluted in reverse osmosis water] at 16°C. Prior to dissection, each animal was anesthetized in a bed of crushed ice and later immersed in a dissecting dish filled with cold leech saline, which contained (in mM) 115 NaCl, 4 KCl, 1.7 CaCl$_2$, 10 D-glucose, and 10 HEPES; pH adjusted to 7.4 with 1 M with NaOH. The animal was then pinned dorsal side up and an incision was made through its dorsal body wall to expose its internal organs. Individual ganglia at segments 3 and 4 were removed and pinned, ventral side up, in individual sylgard-coated 35 × 10 mm petri dishes with pins made from 0.05 mm tungsten wire (California Fine Wire Company, Grover Beach, CA). Each sylgard-coated dish held 1.5 mL of solution and served as a recording chamber in all experiments. Once isolated, the ventral side of each ganglion was then desheathed and continuously superfused with saline at a flow rate of 3 mL/min at room temperature.

### Electrophysiological recordings

Two procedures were used to record the activity of both left and right oscillator heart interneurons from isolated ganglia. In the first procedure, both heart interneurons were recorded simultaneously using two extracellular electrodes. These electrodes were fabricated from microfilament-containing thin-wall capillary glass (o.d. 1 mm, i.d. 0.75 mm; A-M Systems, Sequim, WA) pulled on a Sutter P-97 puller (Sutter Instrument Company, Novato, CA) to have a tip diameter of ~15–20 µm. Each electrode was positioned over the cell body of a neuron and a gentle suction was applied until the cell body was inside the electrode. The extracellular signals were amplified using a differential AC Amplifier Model 1700 (A-M Systems, Sequim, WA) and a Brownlee Precision Model 410 amplifier (AutoMate Scientific, Berkeley, California). Oscillator heart interneurons were identified based on the pattern of their bursting activities and the position of their cell bodies within the ganglion.

In the second procedure, combined intracellular and extracellular recording techniques were used to monitor the activity of both heart interneurons (*Tobin and Calabrese, 2005*). Sharp intracellular electrodes were fabricated from the same glass as the extracellular electrodes and had a resistance of 20–29 MΩ when filled with 2 M C$_2$H$_3$KO$_2$ (KAcetate) and 20 mM KCl. Air bubbles were removed from the tip of each electrode using negative pressure (*Kueh and Jellies, 2012*). In

experiments designed to introduce intracellular leakage of $Na^+$ from an intracellular electrode, the intracellular solution was substituted with 2 M $C_2H_3NaO_2$ (NaAcetate) and 20 mM NaCl, which gave the intracellular electrodes a resistance of 30–40 MΩ.

All intracellular traces were acquired at a sample rate of ≥4 KHz using an Axoclamp 2A amplifier (Molecular Devices, Sunnyvale, CA) in discontinuous current clamp mode or in discontinuous single electrode voltage-clamp mode. During the voltage-clamp experiments, the neuron was voltage-clamped at a holding potential of −45 mV, with a gain set to 0.8 nA/mV, the time constant to 20 ms, and the anti-alias filter to 5 µs. The output bandwidth for intracellular signals was set to 0.3 Khz. For intracellular traces to be accepted into analysis, the input resistance of a neuron that was hyperpolarized with a −0.1 nA pulse had to be ≥60 MΩ and the bath potential at the end of an experiment had to be within ± 5 mV. Both intracellular and extracellular signals were digitized using an Axon Digidata 1440A digitizer and recorded with the Clampex 10.4 software (Molecular Devices, Sunnyvale, CA).

## Solutions

The principal approach used to stimulate the $Na^+/K^+$ pump was to treat the oscillator heart interneurons with 10 µM monensin. Monensin sodium salt (Sigma-Aldrich, St. Louis, MO) was initially diluted in 100% ethanol to prepare a 50 mM stock solution, which was then stored at −20°C. The stock solution was further diluted to 10 µM in saline. To inhibit the $Na^+/K^+$ pump, we would either use a saline that contained 100 µM strophanthidin (Sigma-Aldrich, St. Louis, MO) and that contained 0 mM of $K^+$. Saline solutions with lower concentrations (0.1, 0.4, and 2 mM) of $K^+$ were also used. To control for the effects of ethanol in monensin saline (0.0193% ethanol) or strophanthidin saline (0.217% ethanol), the control saline was supplemented with the same concentration of ethanol.

To block $Ca^{2+}$ and synaptic currents, we replaced the $Ca^{2+}$ in the saline with 1.8 mM $Mn^{2+}$. To block the *h*-current, we added 2 mM $Cs^+$ to the saline solution (*Angstadt and Calabrese, 1989*). To record oscillator heart interneurons as isolated bursters, the ganglia were bathed in normal saline containing 500 µM bicuculline methiodide to block synaptic transmission (Sigma-Aldrich) (*Schmidt and Calabrese, 1992*; *Cymbalyuk et al., 2002*). All solutions were superfused at a rate of 3 mL/min and it took approximately 20 s to replace the entire bathing solution.

## Experimental data analysis

To analyze the burst characteristics of the oscillator heart interneurons, we used custom MATLAB scripts to identify and measure individual bursts based on the methods of *Masino and Calabrese (2002)*. A burst was defined as a group of five or more action potentials, with the burst duration defined as the time between the first and last action potential of each burst. An action potential was detected whenever a voltage change exceeds an inner threshold (approximately 50% of the largest spike amplitude) but not an outer threshold. A refractory period of 20 ms was imposed immediately after the first detection of an action potential to prevent repeated detection of the same action potential. To discriminate between bursts, an interburst interval of 800 ms between each consecutive burst had to elapse before detection of the first action potential from the next burst could occur. Whenever appropriate, these settings were adjusted to ensure that the sample recordings were analyzable by our MATLAB scripts.

We required a minimum of ten consecutive bursts that could be detected for a sample recording to be accepted into analysis. Based on these detected bursts, we were able to analyze five burst characteristics: period, burst duration, interburst interval, duty cycle, and intraburst spike frequency across different experimental treatments (e.g., control vs. monensin). Period was defined as the time from the middle action potential of one burst to the middle action potential of the next burst (*Kristan and Calabrese, 1976*). The middle action potential was chosen because it has more physiological relevance (*Wenning et al., 2014*) and there is less variability in measuring the period based on the middle action potential rather than the first action potential. The burst duration was defined as the time from the first action potential to the last action potential of a burst, and the interburst interval was calculated by subtracting the burst duration from the period. The duty cycle was defined as follows:

$$D = \frac{BD}{T} \times 100\% \tag{1}$$

In *Equation 1*, *D* is the duty cycle, *BD* is the burst duration, and *T* is the period. Because each preparation contributed ≥10 bursts for each experimental treatment, an average was taken for each of the five burst characteristics, resulting in each preparation contributing only one value for each experimental treatment. Moreover, because we did not find any statistically significant differences between the two oscillator heart interneurons of a pair with respect to their burst characteristics, comparisons of burst characteristics in different saline solutions were based on average values taken from both neurons in each ganglion.

In addition to the burst characteristics, we measured the membrane current, membrane conductance, and base potential in experiments involving intracellular recordings. The base potential of an intracellularly recorded oscillator heart interneuron was calculated as follows:

$$t_{BP} = t_{UP} + \frac{(t_T - t_{UP})}{2} \tag{2}$$

In *Equation 2*, $t_{BP}$ is the time point of the base potential, which was defined as midway between the time point of an undershoot phase ($t_{UP}$) and the time point of the next threshold ($t_T$) (*Olsen and Calabrese, 1996*). The undershoot phase was identified based on the time point of the undershoot trough of an action potential. The threshold was identified by taking the third derivative of an action potential and identifying the time point of the first positive peak of that third derivative (*Henze and Buzsáki, 2001*). In cases where spiking activity had been suppressed for a minute or more, the base potential would be the resting membrane potential at any given time point during that period of suppressed spiking activity. Measurements of specific time points, membrane potentials, and currents were performed using Clampfit 10 (Molecular Devices, Sunnyvale, CA) and LabChart Reader 8 (ADInstruments, Colorado Springs, CO).

The experimental data from within-group experiments (e.g., periods from the same preparations in two or more saline solutions) were analyzed using a paired t-test or a repeated-measures ANOVA, depending on the number of experimental treatments being compared. In between-groups experiments (e.g., 1 vs 10 µM monensin), an unpaired t-test or an ANOVA was used instead. Finally, in the $Na^+$ and $K^+$ experiment, which has time as the within-group factor and electrode cation ($Na^+$ and $K^+$) as the between-group factor, a split-plot ANOVA was used to analyze changes in base potential from both groups over time. Whenever an ANOVA revealed a statistical significance, a Tukey's range test or a Holm-Šídák test was used to identify specific sample means that were significantly different. Statistical significance was defined as $p < 0.05$ for all tests. All statistical analyses were done using SigmaPlot 12 (Systat Software, Inc., San Jose, CA) and IBM SPSS statistics 12 (IBM Corp., Armonk, NY). All experimental data are represented as mean ± SEM and were plotted using Microsoft Excel 2016 (Microsoft Corp, Redmond, WA). All figures were prepared using Adobe Illustrator CS5 (Adobe Systems Inc., San Jose, CA).

## Mathematical model

We developed a single-compartment model of a heart interneuron using Hodgkin-Huxley style equations. Our model has a leak ($I_{leak}$) current and a $Na^+/K^+$ pump ($I_{pump}$) current, with the leak current having $Na^+$ ($I_{(leak,Na)}$) and $K^+$ ($I_{(leak,K)}$) components. The model also has eight voltage-gated currents: a fast $Na^+$ current ($I_{Na}$), a persistent $Na^+$ current ($I_P$), a low-threshold rapidly inactivating $Ca^{2+}$ current ($I_{CaF}$), a low-threshold slowly inactivating $Ca^{2+}$ current ($I_{CaS}$), a hyperpolarization-activated inward current ($I_h$), a delayed rectifier-like $K^+$ current ($I_{K1}$), a persistent $K^+$ current ($I_{K2}$), and a fast transient $K^+$ current ($I_{KA}$). The model of a single heart interneuron can be converted into a half-center oscillator by including a spike-mediated synaptic current ($I_{SynS}$) and a graded synaptic current ($I_{SynG}$), as follows:

$$C\frac{dV}{dt} = -\left(I_{Na} + I_P + I_{K1} + I_{K2} + I_{KA} + I_{h,Na} + I_{h,K} + I_{CaF} + I_{CaS} + I_{Leak,Na} + I_{Leak,K} + I_{Pump} + I_{SynS} + I_{SynG}\right) \tag{3}$$

where *C* is the membrane capacitance (in nF), *V* is the membrane potential (in *V*), *t* is time (in s). Our half-center oscillator model in *Equation 3* differs from the original *Hill et al. (2001)* model because it includes a $Na^+/K^+$ pump current and it describes changes in intracellular $Na^+$ concentrations that occur as a result of the $Na^+$ fluxes carried by ionic currents, $Na^+/K^+$ pumps, and monensin-facilitated diffusion:

$$\frac{\mathrm{d}[Na]_i}{\mathrm{d}t} = M\big([Na]_o - [Na]_i\big) - \frac{I_P + I_{Na} + I_{h,Na} + I_{leak,Na} + 3\,I_{pump}}{vF} \tag{4}$$

In **Equation 4**, $[Na]_i$ is the changing intracellular Na$^+$ concentration, $[Na]_o$ is the extracellular Na$^+$ concentration that was kept constant, v is the volume (~6.7 pL) of the intracellular Na$^+$ reservoir, F is Faraday's constant, and $M$ is the exchange rate (in 1/s) of Na$^+$ and H$^+$ by monensin, which is based on Fick's Law of diffusion. Because the Na$^+$/K$^+$ pump exchanges two K$^+$ ions for three Na$^+$ ions, the contribution of the pump current to intracellular Na$^+$ concentrations was multiplied by a factor of 3. The Na$^+$/K$^+$ pump current has a sigmoidal dependence on intracellular Na$^+$ concentrations, which is expressed as follows:

$$I_{pump} = \frac{I_{pump}^{max}}{1 + exp\left(\frac{[Na]_{ih} - [Na]_i}{[Na]_{is}}\right)} \tag{5}$$

where $I_{pump}^{max}$ is the maximum Na$^+$/K$^+$ pump current, $[Na]_{ih}$ is the intracellular Na$^+$ concentration for the half-activation of the Na$^+$/K$^+$ pump, and $[Na]_{is}$ the sensitivity of the Na$^+$/K$^+$ pump to $[Na]_i$.

The h-current and the leak current have Na$^+$ and K$^+$ components. In the case of the h-current, we computed the Na$^+$ component ($I_{h,Na}$) using the equilibrium potential of Na$^+$ and we computed the K$^+$ component ($I_{h,K}$) using the equilibrium potential of K$^+$:

$$I_{h,Na} = \frac{3}{7}\bar{g}_h m_h^2 (V_m - E_{Na}) \tag{6}$$

$$I_{h,K} = \frac{4}{7}\bar{g}_h m_h^2 (V_m - E_K) \tag{7}$$

In both equations, $\bar{g}_h$ is the maximum conductance, $m_h$ is the activation variable, $V_m$ is the membrane potential, and $E_{ion}$ is the equilibrium potential at 20°C. Unlike the equilibrium potential of K$^+$, the equilibrium potential of Na$^+$ was computed at each time step as a function of a constant extracellular Na$^+$ concentration and a changing intracellular Na$^+$ concentration:

$$E_{Na} = 0.02526 \ln\left(\frac{[Na]_o}{[Na]_i}\right) \tag{8}$$

We also computed two components ($I_{leak,Na}$ and $I_{(leak,K)}$ of the leak current:

$$I_{leak,Na} = \bar{g}_{leak,Na}(V_m - E_{Na}) \tag{9}$$

$$I_{leak,K} = \bar{g}_{leak,K}(V_m - E_K) \tag{10}$$

We fixed the ratio of Na$^+$ to K$^+$ conducted by the leak current and used the equilibrium potentials of K$^+$ and Na$^+$ from **Hill et al. (2001)** to compute this ratio. The equations for $\bar{g}_{leak,Na}$ and $\bar{g}_{leak,K}$ are:

$$\bar{g}_{leak,Na} = \bar{g}_{leak}\frac{\big(E_{leak,Ref} - E_K\big)}{\big(E_{Na,Ref} - E_K\big)} \tag{11}$$

$$\bar{g}_{leak,K} = \bar{g}_{leak}\frac{\big(E_{leak,Ref} - E_{Na,Ref}\big)}{\big(E_K - E_{Na,Ref}\big)} \tag{12}$$

In **Equation 12**, $E_{Na,Ref}$ was fixed at 0.045 V. The Na$^+$ and K$^+$ components of the leak conductance, $\bar{g}_{leak,Na}$ and $\bar{g}_{leak,K}$, were computed ahead of time and were fixed for the duration of each simulation.

We computed the average value function $I$ of the Na$^+$/K$^+$ pump current and the h-current over a bounded integral as follows:

$$I = \frac{1}{t_2 - t_1}\int_{t_1}^{t_2} I\,dt \tag{13}$$

Both average currents were computed over each burst duration and over each interburst interval. When the average currents were computed over a burst duration, $t_1$ represented the time of the first action potential in a burst whereas $t_2$ represented the time of the last action potential in the same burst. When the average currents were computed during an interburst interval, in which the membrane potential was below $-50$ mV, $t_1$ represented the time at which the membrane potential crossed $-50$ mV after the last action potential in the preceding burst whereas $t_2$ represented the time at which the membrane potential crossed $-50$ mV immediately before the first action potential of the next burst. We used the trapezoid method for numerical integration (the trapz function in MATLAB).

A complete list of Hodgkin-Huxley style equations that describe ionic and synaptic currents can be found in the Appendix. We computed numerical solutions to these ordinary differential equations using the 8–9 order Prince-Dormand method from the GNU Scientific Library (www.gnu.org/software/gsl). All variables were computed with an absolute tolerance of $1e^{-9}$s, a relative tolerance of $1e^{-10}$s, and a maximum time step of $1e^{-3}$s.

We analyzed the burst characteristics of current and voltage trajectories from our model with custom-made routines written in the C programming language. The four burst characteristics that we measured were the period, burst duration, interburst interval, and duty cycle. The burst characteristics from our model were measured in the same way as the burst characteristics from our experimental data (see Experimental data analysis).

# Additional information

### Competing interests
RLC: Reviewing editor, *eLife*. The other authors declare that no competing interests exist.

### Funding

| Funder | Grant reference number | Author |
| --- | --- | --- |
| National Institute of Neurological Disorders and Stroke | R01 NS085006 | Ronald L Calabrese |
| National Science Foundation | PHY-0750456 | Gennady S Cymbalyuk |

The funders had no role in study design, data collection and interpretation, or the decision to submit the work for publication.

### Author contributions
DK, Performed the experiments and analyzed the experimental data; WHB, Wrote the programs and performed the computer simulations; GSC, RLC, Supervised the project and provided valuable input

### Author ORCIDs
Daniel Kueh, http://orcid.org/0000-0002-2462-5203
Ronald L Calabrese, http://orcid.org/0000-0001-7135-3469

# Additional files

### Major datasets
The following dataset was generated:

| Author(s) | Year | Dataset title | Dataset URL | Database, license, and accessibility information |
| --- | --- | --- | --- | --- |
| Daniel Kueh, William H Barnett, Gennady S Cymbalyuk, Ronald L Calabrese | 2016 | Data from: Na+ K+ pump interacts with the h-current to control bursting activity in central pattern generator neurons of leeches | http://dx.doi.org/10.5061/dryad.fr1r2 | Available at Dryad Digital Repository under a CC0 Public Domain Dedication |

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

## Appendix

The current balance equation takes the form:

$$C\frac{\mathrm{d}V}{\mathrm{d}t} = -\left(I_{\mathrm{Na}} + I_{\mathrm{P}} + I_{\mathrm{K1}} + I_{\mathrm{K2}} + I_{\mathrm{KA}} + I_{\mathrm{h,Na}} + I_{\mathrm{h,K}} + I_{\mathrm{CaF}} + I_{\mathrm{CaS}} + I_{\mathrm{Leak,Na}} + I_{\mathrm{Leak,K}} + I_{\mathrm{Pump}} + I_{\mathrm{SynS}} + I_{\mathrm{SynG}}\right)$$

$$I_{\mathrm{pump}} = \frac{I_{\mathrm{pump}}^{\mathrm{max}}}{1 + exp\left(\frac{[\mathrm{Na}]_{\mathrm{ih}} - [\mathrm{Na}]_{\mathrm{i}}}{[\mathrm{Na}]_{\mathrm{is}}}\right)},$$

$$I_{\mathrm{Na}} = \bar{g}_{\mathrm{Na}} m_{\mathrm{Na}}^3 h_{\mathrm{Na}} (V - E_{\mathrm{Na}}),$$

$$I_{\mathrm{P}} = \bar{g}_{\mathrm{P}} m_{\mathrm{P}} (V - E_{\mathrm{Na}}),$$

$$I_{\mathrm{K1}} = \bar{g}_{\mathrm{K1}} m_{\mathrm{K1}}^2 h_{\mathrm{K1}} (V - E_{\mathrm{K}}),$$

$$I_{\mathrm{K2}} = \bar{g}_{\mathrm{K2}} m_{\mathrm{K2}}^2 (V - E_{\mathrm{K}}),$$

$$I_{\mathrm{KA}} = \bar{g}_{\mathrm{KA}} m_{\mathrm{KA}}^2 h_{\mathrm{KA}} (V - E_{\mathrm{K}}),$$

$$I_{\mathrm{h,Na}} = \frac{3}{7} \bar{g}_{\mathrm{h}} m_{\mathrm{h}}^2 (V_m - E_{\mathrm{Na}}),$$

$$I_{\mathrm{h,K}} = \frac{4}{7} \bar{g}_{\mathrm{h}} m_{\mathrm{h}}^2 (V_m - E_{\mathrm{K}}),$$

$$I_{\mathrm{CaF}} = \bar{g}_{\mathrm{CaF}} m_{\mathrm{CaF}}^2 h_{\mathrm{CaF}} (V - E_{\mathrm{Ca}}),$$

$$I_{\mathrm{CaS}} = \bar{g}_{\mathrm{CaS}} m_{\mathrm{CaS}}^2 h_{\mathrm{CaS}} (V - E_{\mathrm{Ca}}),$$

$$I_{\mathrm{leak,Na}} = \bar{g}_{\mathrm{leak,Na}} (V - E_{\mathrm{Na}}),$$

$$I_{\mathrm{leak,K}} = \bar{g}_{\mathrm{leak,K}} (V - E_{\mathrm{K}}),$$

The expressions for $\bar{g}_{\mathrm{leak,Na}}$ and $\bar{g}_{\mathrm{Leak,K}}$ are

$$\bar{g}_{\mathrm{leak,Na}} = \bar{g}_{\mathrm{leak}} \frac{(E_{\mathrm{leak,Ref}} - E_{\mathrm{K}})}{(E_{\mathrm{Na,Ref}} - E_{\mathrm{K}})} \text{ and}$$

$$\bar{g}_{\mathrm{leak,K}} = \bar{g}_{\mathrm{leak}} \frac{(E_{\mathrm{leak,Ref}} - E_{\mathrm{Na,Ref}})}{(E_{\mathrm{K}} - E_{\mathrm{Na,Ref}})}.$$

Parameters take values $I_{\text{pump}}^{\max}$ = 0.4 nA, $[\text{Na}]_{\text{o}}$= 0.115 M, $[\text{Na}]_{\text{is}}$= 0.0004 M, $[\text{Na}]_{\text{ih}}$= 0.0145 M, $C$ = 0.5 nF, $\bar{g}_{\text{Na}}$ = 200 nS, $\bar{g}_{\text{P}}$= 2.36 nS, $\bar{g}_{\text{K1}}$=100 nS, $\bar{g}_{\text{K2}}$= 119 nS, $\bar{g}_{\text{KA}}$= 80 nS, $\bar{g}_{\text{h}}$= 4.89 nS, $\bar{g}_{\text{CaF}}$= 17 nS, $\bar{g}_{\text{CaS}}$ = 3.38 nS, $\bar{g}_{\text{leak}}$ = 9.09 nS, $E_{\text{leak,Ref}}$ = − 0.055 V, $E_{\text{Na,Ref}}$ = 0.045 V, $E_{\text{K}}$ = −0.07 V, and $E_{\text{Ca}}$ = 0.135 V.

The monensin rate constant was given a non-zero value to represent the application of monensin in two different experimental treatments: $M$ = 2.2125 × 10$^{-3}$ /s for monensin saline and $M$ = 1.875 × 10$^{-4}$ /s for monensin plus Cs$^{+}$ saline.

The activation and inactivation of the endogenous currents are governed by the following expressions:

$$\frac{dm_{\text{K2}}}{dt} = \frac{f_\infty(-83,\ 0.01748, V) - m_{\text{K2}}}{\tau(200,\ 0.035,\ 0.057,\ 0.043, V)}$$

$$\frac{dm_{\text{P}}}{dt} = \frac{f_\infty(-120,\ 0.04761, V) - m_{\text{P}}}{\tau(400,\ 0.057,\ 0.01,\ 0.2, V)}$$

$$\frac{dm_{\text{Na}}}{dt} = \frac{f_\infty(-150,\ 0.029,\ V) - m_{\text{Na}}}{0.0001}$$

$$\frac{dh_{\text{Na}}}{dt} = \frac{f_\infty(500,\ 0.030, V) - h_{\text{Na}}}{\tau_{hNa}(V)}$$

$$\frac{dm_{\text{CaF}}}{dt} = \frac{f_\infty(-600,\ 0.0487, V) - m_{\text{CaF}}}{\tau_{m\text{CaF}}(V)}$$

$$\frac{dh_{\text{CaF}}}{dt} = \frac{f_\infty(350,\ 0.0515, V) - h_{\text{CaF}}}{\tau(270,\ 0.055,\ 0.06,\ 0.31, V)}$$

$$\frac{dm_{\text{CaS}}}{dt} = \frac{f_\infty(-420,\ 0.0482,\ V) - m_{\text{CaS}}}{\tau(-400,\ 0.0487,\ 0.005,\ 0.134, V)}$$

$$\frac{dh_{\text{CaS}}}{dt} = \frac{f_\infty(216,\ 0.0607, V) - h_{\text{CaS}}}{\tau(-250,\ 0.043,\ 0.2,\ 5.25, V)}$$

$$\frac{dm_{\text{K1}}}{dt} = \frac{f_\infty(-143,\ 0.021, V) - m_{\text{K1}}}{\tau(150,\ 0.016,\ 0.001,\ 0.011, V)}$$

$$\frac{dh_{\text{K1}}}{dt} = \frac{f_\infty(111,\ 0.028, V) - h_{\text{K1}}}{\tau(-143,\ 0.013,\ 0.5,\ 0.2, V)}$$

$$\frac{dm_{\text{KA}}}{dt} = \frac{f_\infty(-130,\ 0.044, V) - m_{\text{KA}}}{\tau(200,\ 0.03,\ 0.005,\ 0.011, V)}$$

$$\frac{dh_{KA}}{dt} = \frac{f_\infty(160,\ 0.063,\ V) - h_{KA}}{\tau(-300,\ 0.055,\ 0.026,\ 0.0085,\ V)}$$

$$\frac{dm_h}{dt} = \frac{f_{h\infty}(V) - m_h}{\tau(-100,\ 0.073,\ 0.7,\ 1.7,\ V)}$$

The steady-state activation ($f_\infty$) of most variables is:

$$f_\infty(a,b,V) = \frac{1}{1 + exp[a(V+b)]}$$

and the time constant ($\tau$) takes the form:

$$\tau(a,b,c,d,V) = c + \frac{d}{1 + \exp[a(V+b)]}$$

For the gating variables, $h_{Na}$, $m_h$, and $m_{CaF}$, we used the following expressions:

$$f_{h\infty}(V) = \frac{1}{1 + 2\exp[1.023 \times 180(V+0.049)] + \exp[1.023 \times 500(V+0.049)]},$$

$$\tau_{hNa}(V) = 0.004 + \frac{0.006}{1 + \exp[500(V\ +\ 0.028)]} + \frac{0.01}{cosh[300(V\ +\ 0.027)]},\ \text{and}$$

$$\tau_{mCaF}(V) = 0.011 + \frac{0.024}{cosh[-330(V\ +\ 0.0467)]}$$

We modeled the intracellular Na$^+$ by taking into account the flux of Na$^+$ through ionic currents, the Na$^+$/K$^+$ pump, and diffusion:

$$\frac{d[Na]_i}{dt} = M\big([Na]_o - [Na]_i\big) - \frac{I_P + I_{Na} + I_{h,Na} + I_{leak,Na} + 3I_{pump}}{vF}$$

In simulations, we gave the value of 650 for the term vF. When modeling the heart interneurons as a half-center oscillator, we included two synaptic currents for each neuron:

$$I_{SynS} = \bar{g}_{SynS} Y_{post} M_{post}\big(V_{Post} - E_{Syn}\big)\ \text{and}$$

$$I_{SynG} = \bar{g}_{SynG} \frac{P_{Pre}^3}{C + P_{Pre}^3}\big(V_{Post} - E_{Syn}\big)$$

The synaptic conductance parameters were $\bar{g}_{SynS}$ = 37 nS, $\bar{g}_{SynG}$ = 12.8 nS, and $E_{Syn}$ = – 0.07 V. Spike-mediated synaptic activation was determined by the following expressions:

$$\frac{dX_{Post}}{dt} = \frac{X_\infty(V_{Pre}) - X_{Post}}{\tau_1},$$

$$\frac{dY_{Post}}{dt} = \frac{X_{Post} - Y_{Post}}{\tau_2} \text{ , and}$$

$$\frac{dM_{Post}}{dt} = \frac{M_\infty(V_{Pre}) - M_{Post}}{\tau_3}.$$

The steady-state functions for the spike-mediated synapses took the following form:

$$X_\infty(V_{Pre}) = \frac{1}{1 + \exp[-1738(V_{Pre} + 0.01)]}$$

$$M_\infty(V_{Pre}) = 0.1 + \frac{0.9}{1 + \exp[-1000(V_{Pre} + 0.04)]}$$

The time constants were $\tau_1$ = 0.002 s, $\tau_2$ = 0.011 s, and $\tau_3$ = 0.2 s.

The graded synapses were computed based on presynaptic $Ca^{2+}$ concentration:

$$\frac{dP_{Pre}}{dt} = I_{Ca,Pre} - BP_{Pre}$$

where $B$ is the buffering rate constant, which takes value 10 s$^{-1}$.

$$I_{Ca,Pre} = \max(0, -I_{CaF} - I_{CaS} - A).$$

$$\frac{dA_{Post}}{dt} = \frac{A_\infty(X_{Pre}) - X_{Post}}{0.2}$$

$$A_\infty(V_{Pre}) = \frac{10^{-10}}{1 + exp[-100(V_{Pre} + 0.02)]}$$

