## [Decision Letter]

Thank you for resubmitting your work entitled "Na+/K^+^ pump interacts with the *h*-current to control bursting activity in central pattern generator neurons of leeches" for further consideration at *eLife*. Your revised article has been favorably evaluated by Eve Marder (Senior Editor), Leslie Griffith (Reviewing Editor), and two Reviewers.

The manuscript has been greatly improved but there are some remaining issues that need to be addressed before acceptance, as outlined below:

In recent years the role of Na+/K^+^ ATPase in modulating neural circuit output has been investigated in many different model systems. Its fundamental role of maintaining intracellular ion concentration has made it very difficult to manipulate without affecting other electrical properties. This study, through carefully controlled pharmacology buttressed with a theoretical model, provides strong evidence that an interaction between the hyperpolarizing pump current generated by the Na+/K^+^ ATPase and the hyperpolarization-activated inward current Ih has a substantial influence on the half-center oscillations of leech heartbeat interneurons. The findings are of broad interest as this intrinsic self-regulation may be present in other neuronal circuits and play a similar role.

The paper is much improved over its previous version but the reviewers felt that there were still a few issues where more discussion was merited or interpretations might need to be softened. Specifically:

1) The bicuculline+Cs+monesin results need to be more carefully discussed. The authors say "Adding monensin to the bicuculline and Cs+ saline did not significantly change any of the burst characteristics relative to bicuculline and Cs+ saline." Why does monensin have no effect at all when bicuculline is present? This is a bit surprising and should be discussed.

We have clarified these points by altering the paragraph in question as follows:

“Similar results were also observed when we stimulated the pump of oscillator heart interneurons that have been isolated with 500 μM bicuculline, which blocks the synaptic transmission between the two oscillator heart interneurons (Figure 4—figure supplement 1) (Schmidt and Calabrese, 1992; Cymbalyuk et al., 2002).

Although these oscillator heart interneurons form half-center oscillators, they can function as endogenous bursters when recorded extracellularly in bicuculline saline (Cymbalyuk et al., 2002). The one notable exception was that stimulating the pump with monensin in isolated oscillator heart interneurons does not significantly decrease the period (3.9 ± 0.1 s for bicuculline saline vs. 3.3 ± 0.1 s for bicuculline plus monensin saline, n = 5), which can be explained in part by a floor effect; period had already been significantly reduced in bicuculline saline (8.8 ± 0.9 s for control saline vs. 3.9 ± 0.1 s for bicuculline saline, n = 5, Tukey’s test, p < 0.001), to a point where adding monensin could not have a significant effect on shortening period (Figure 4—figure supplement 1). When the h-current was blocked, the period increased significantly relative to bicuculline saline (3.5 ± 0.1 s for bicuculline saline vs. 7.1 ± 0.9 s for bicuculline plus Cs+ saline, n = 5, Tukey’s test, p = 0.003). Likewise, the interburst interval increased significantly as 2 well relative to bicuculline saline (1.3 ± 0.05 s for bicuculline saline vs. 5.8 ± 1.0 s for bicuculline plus Cs+ saline, n = 5, Tukey’s test, p = 0.0004). The burst duration, however, decreased significantly relative to bicuculline saline (2.2 ± 0.1 s for bicuculline saline vs. 1.3 ± 0.1 s for bicuculline plus Cs+ saline, n = 5, Tukey’s test, p = 0.047). Adding monensin to the bicuculline plus Cs+ saline did not significantly change any of the burst characteristics relative to bicuculline plus Cs+ saline

(Figure 4—figure supplement 1). Thus, h-current must be present for monensin to decrease period of isolated heart interneurons (bicuculline).”

2) The h-current can be clearly observed in Figure 5 when membrane potential is hyperpolarized, whereas it is totally absent in the presence of strophanthidin (Figure 5). Any explanations? It is odd that in the presence of strophanthidin there is no sag potential.

[Editors’ note: a previous version of this study was rejected after peer review, but the authors submitted for reconsideration. The first decision letter after peer review is shown below.]

Thank you for choosing to send your work entitled "The Na+/K^+^ pump interacts with the h-current to control burst period by central pattern generator neurons of leeches" for consideration at *eLife*. Your full submission has been evaluated by Eve Marder (Senior editor) and 4 peer reviewers, one of whom, Leslie Griffith, is a member of our Board of Reviewing Editors, and the decision was reached after many, many discussions between the reviewers. Based on our discussions and the individual reviews below, we regret to inform you that your work in its present form is not suitable for publication in *eLife*. As you can see below, many of the reviewers' problems can be dealt with rewriting, but there was sufficient confusion/disagreement about other points so that reviewers were unable to agree that relatively minor changes would make the paper acceptable, and therefore we are forced to reject it at this time.

The authors present an interesting electrophysiological analysis and modeling study of the mechanisms underlying the effect of Na/K pump current on bursting parameters in leech heart interneurons. They performed an analysis of how the pump current interacts with the h-current to alter the burst period in both half-center oscillators and isolated interneurons. They then test their hypothesis with a model to confirm the plausibility of the findings from their experiments. They conclude that the h-current can be activated by the hyperpolarizing pump current and is important for controlling burst period. These results, if they could be substantiated more strongly would represent an important contribution to our understanding of rhythmic activity.

While all the referees found the paper interesting and potentially important, there was substantial concern about the interpretation of the experiments and the strength of the conclusions that the manuscript makes based on the techniques used. There was also strong agreement among the reviewers and the editors that the paper as presented was incredibly difficult to read and poorly put together especially in terms of the descriptions of the figures in the text and figure legends.

The major experimental issue with the paper revolves around the ability to interpret some of the manipulations. The authors would like to be able to conclude that only the contribution of the dynamics of the pump with respect to rhythmic activity is being altered by their conditions. The concern is that manipulating the pump pharmacologically has the fundamental problem that it alters both the dynamics of the pump and its function in stabilizing membrane potentials.

In addition, there are concerns about the specificity of the manipulations. Monensin has not been validated in the leech system and the fact that it acts by increasing Na^+^ levels leaves open many possibilities of how it could affect cellular function. Validation by intracellular recording and perhaps showing that its actions are blocked by ouabain would be helpful in establishing its mechanism. Likewise, the use of zero K^+^ is problematic since this manipulation would affect many processes. Because of the reliance on the extracellular recordings, some of the conclusions are less direct than they need to be, in terms of how membrane potential or K^+^ currents are being affected.

The reviewers also did not think that the modeling component of the paper was very useful. Exploring the parameters of the pump dynamics more fully instead of simply showing that the cell could be working as posited would be more rigorous and generally informative.

Reviewer #1:

This manuscript sets out to determine the role of the Na+/K^+^ ATPase (pump) in regulating bursting activity in the neural network that controls leech heartbeat. The pump is electrogenic and strong activation by Na influx during spiking can lead to hyperpolarization. The authors use electrophysiology and pharmacology to determine the effect of the pump on interneurons oscillating either in halfcenter mode or in synaptic isolation, and then do some proof-of-concept computational modeling. The main claim here is that the interaction between pump currents and hyperpolarization-activated (h^-^) currents is essential for normal bursting behavior.

The effect of hyperpolarization and its interaction with h-currents has been studied in different contexts, but the potential role for network dynamics has been mostly overlooked. The topic is therefore of potential significance for many systems. However, I found a number of flaws in this study.

Although the writing is mostly clear, I find the narrative could be improved. This is mostly due to the sequence in which the data are presented. The paper falls into three sections. First, the effect of monensin on bursting activity and its interaction with h-currents is described. Then follows a set of experiments aimed at providing evidence for the specificity of monensin effects by using intracellular electrophysiology while manipulating the pump in different ways. Last, the modeling work is presented to address the mechanisms underlying the effects on neuronal activity. In my opinion, the flow of this work could be much improved by addressing the cellular actions of monensin first, providing evidence that it is really the pump that is activated. The effect on bursting activity can then be addressed experimentally and theoretically without interruption. In addition, results described in the text and presented in the figures should be in the same sequence, which is not always the case.

For a number of reasons it is difficult to show that the effects on network activity are really due to selective activation of the pump. For this reason, the authors show a number of approaches to provide parallel lines of evidence. While the voltage-clamp experiments showing that monensin increases outward current without affecting conductance are fairly convincing, other approaches (AHP measurments, zero K^+^, strophanthidin) and are flawed because they allow other interpretations.

Other than the argument that the computational model can recreate the firing behavior seen in the experiments, it is not critically addressed if monensin effects are really just due to direct action on the interneurons studied. With the interneurons in halfcenter mode, are there other intact inputs? Even in the other set of experiments, doesn't synaptic isolation with bicucullin just mean isolation from each other? There may be good arguments against this being a problem, but for anyone not familiar with the system it isn't evident that the network effects couldn't be due to effects on presynaptic or modulatory neurons.

The computational models replicate the network effects and therefore serve as proof-of-concept, i.e. show that the ionic mechanisms included are sufficient to explain the effects. However, they fall somewhat short in giving a mechanistic explanation of the specific effects. The interactions of pump currents, h-currents, and Ca^2+^ currents are analyzed solely from the model traces replicating the different experimental conditions. The analysis is therefore mostly descriptive and correlative. There is no careful independent manipulation of the critical parameters, and therefore the statements about complex interactions between different currents are mostly hand-waving.

Reviewer #2:

The authors present an interesting electrophysiological analysis and modelling study of the mechanisms underlying the effect of Na/K pump current on bursting parameters in leech heart interneurons. They have performed a detailed analysis of how pump current interact with h-current to alter the burst period in both half-centre oscillators and isolated interneurons. They then test their hypothesis in a modified computer model and confirm the findings from their animal experiments. They provide evidence that monensin induced a sustained outward current and assume it is the pump current. They conclude that the h-current can be activated by hyperpolarizing pump current and is important for controlling burst period. The conclusions are of broad interest to those studying neuronal networks, especially central pattern generators, because they demonstrate a type of general intrinsic self-regulation that may present in many networks, and can be targeted by modulators. This is an interesting and generally well designed work which would benefit from attention to the following issues.

Major concerns:

1) In this study monensin is used to increase Na^+^ concentrations and in turn enhance Na/K pump current. Experiments have been done to demonstrate that the outward current induced by monensin is the pump current, however more direct evidence is still lacking. This probably can be easily provided by adding ouabain/strophanthidin to the voltage clamped interneurons to block the change of holding current. Or using ouabain/strophanthidin in experiments shown in Figure 5.

Although monensin has been used widely to enhance Na/K pump current, it actually unspecifically increase the intracellular Na^+^ concentration, which in turn has effects on various targets, e.g. changing Na^+^ current kinetic and activating Na+-dependent K^+^ channels.

2) Studies have shown that A-type current is also linked to the self-regulation of Na/K pump function, and alter neuron firing, e.g. Pulver and Griffith's paper, which is also cited here. Have the authors considered this possibility in their studies? If this current can be ruled out, it may be worth to mention it in the Discussion.

3) I assume that the dual extracellular recordings with Cs^+^ and monensin (Figure 1) are always in the sequence of Cs+ and then monensin. Have the authors tried the other way around, adding monensin first then Cs+?This is to rule out the possibility that the sequence may affect the result, and confirm the outcome.

The authors compared the burst period, burst duration, and interburst interval of half-center oscillators treated with monensin saline (Figure 2) to those of half-center oscillators treated with Cs+ plus monensin saline (Figure 3), which are unpaired data. If some extra experiments have been done in the sequence suggested, these data can be paired.

4) It would be logical to show the data that demonstrating monensin indeed enhances Na/K pump current in the beginning of the Results, which will be the same sequence as in the Discussion.

Reviewer #3:

This is a very detailed analysis that I found extremely difficult to digest. This is not necessarily a criticism of the paper (but see below for suggested changes). Perhaps it was due to a combination of my not being a leech neurobiologist, plus my own cognitive limitations. I did try to carefully examine every result for every pharmacological manipulation, but frequently the bigger picture became lost in the blizzard of details.

Overall, both the physiology and the modelling seem to be sound, and I have no major concerns about the interpretation of the results. The paper makes a valuable contribution, it was just very hard to read.

The order in which the results are presented caused some aggravation, the most important being that the effect of monesin is assumed without any comment whatsoever to be entirely via its effect on the Na/K pump, rather than some other mechanism involving increased intracellular Na or pH change. This assumption is clearly apparent in the heading of the first section of Results, which asserts that the Na/K pump is being stimulated, and doesn't even mention monesin.

I was greatly relieved when I got to Figure 5 and Figure 6, which presents evidence that the pharmacolocigal agents (10 μM monesin and 2 mM Cs+) used in these experiments are acting via the mechanisms that the authors assume, but by then I had plowed through the detailed analysis of effects of various combinations of monesin, Cs+ and bicuculine on burst period, burst duration, interburst interval, duty cycle and spike frequency, all the while assuming that the authors had not bothered to examine possible alternative pharmacological effects. Some of this problem might be avoided by presenting the results in Figure 5 and Figure 6 before presenting the detailed analysis in Figure 1–Figure 4. This would give the reader more confidence in the interpretation of results.

The authors also need to cite evidence that 2 mM Cs+ is acting via its effect on the h-current, rather than some other K^+^ current. I am hoping that the effects of 2 mM Cs+ have previously been substantiated in these neurons, because this key assumption is not sufficiently supported by the experiments in this paper.

Any way that the authors can find to reorganize the results to make them more easily digested would be helpful.

Reviewer #4:

This manuscript is extremely tantalizing and frustrating at the same time. On the one hand, its conclusions are potentially interesting and important, and the data presented are all plausible. That said, the manuscript itself is unduly long and cumbersome and there are a number of cases where there are missing links in the story.

Essentially the authors wish to show that the Na/K pump is dynamically activated during bursting in leech such that the pump interacts with Ih to help determine oscillator period. This would make a nice and important result.

That said, it almost feels as if there are 2-3 papers here, each of which needs more fleshing out, which are not done justice by putting them together. The theory section seems too fragile to be more than a "it could work like this". Having a model to say "it could work like this" isn't necessarily bad, but is counterproductive unless it is relatively robust to the natural variation in the membrane conductances in the system, which it may or may not be.

The issues in the paper that confound its simple interpretation:

a) The authors use 10uM monensin to putatively activate the pump by increasing the intracellular Na concentration. This is an established drug (but a higher concentration than often used), with known pharmacological properties in other systems, but we are not given any indications that the authors have validated its use in leech, except to show that its effects are similar to direct Na injection. It is not totally obvious what the validation could be, but minimally, some kind of dose response using intracellular recordings to see that the change in membrane potential is a graded function of drug concentration. And that the effects are counteracted by oubain, in a dose-dependent manner? Would that be helpful? And were these high concentrations necessary?

b) Using zero extracellular K is a strange thing to do, as that means the K equilibrium potential is now ill-defined. It would be far better to do normal, 1/2, 1/10 K, so that it would be possible to interpret the results of low K on the outward currents.

c) The modeling section is unsatisfying in that it tries to do too much and therefore it does too little. For example, it would be interesting to see the effect of changes of the monensin rate on the behavior of the isolated single neuron or the half-center. As it stands, it sounds like the monensin rate was changed for different simulations? Did I misunderstand this? And it would be great to see the effect of different moninsen rates on neurons with different Ih's. Did I miss this? If the model is as sensitive as it seems that it might be, what does this mean for the biological system? And wouldn't that have necessitated matching with different drug concentrations? This is all confusing.

---

## [Author Response]

*Thank you for resubmitting your work entitled "Na+/K^+^ pump interacts with the h-current to control bursting activity in central pattern generator neurons of leeches" for further consideration at eLife. Your revised article has been favorably evaluated by Eve Marder (Senior Editor), Leslie Griffith, a Reviewing Editor, and two Reviewers.*

*The manuscript has been greatly improved but there are some remaining issues that need to be addressed before acceptance, as outlined below:*

*In recent years the role of Na+/K^+^ ATPase in modulating neural circuit output has been investigated in many different model systems. Its fundamental role of maintaining intracellular ion concentration has made it very difficult to manipulate without affecting other electrical properties. This study, through carefully controlled pharmacology buttressed with a theoretical model, provides strong evidence that an interaction between the hyperpolarizing pump current generated by the Na+/K^+^ ATPase and the hyperpolarization-activated inward current Ih has a substantial influence on the half-center oscillations of leech heartbeat interneurons. The findings are of broad interest as this intrinsic self-regulation may be present in other neuronal circuits and play a similar role.*

*The paper is much improved over its previous version but the reviewers felt that there were still a few issues where more discussion was merited or interpretations might need to be softened. Specifically:*

*1) The bicuculline+Cs+monesin results need to be more carefully discussed. L361-362 says "Adding monensin to the bicuculline and Cs+ saline did not significantly change any of the burst characteristics relative to bicuculline and Cs+ saline." Why does monensin have no effect at all when bicuculline is present? This is a bit surprising and should be discussed.*

We have clarified these points by altering the paragraph in question as follows:

“Similar results were also observed when we stimulated the pump of oscillator heart interneurons that have been isolated with 500 μM bicuculline, which blocks the synaptic transmission between the two oscillator heart interneurons (Figure 4—figure supplement 1) (Schmidt and Calabrese, 1992; Cymbalyuk et al., 2002).

[…]

Thus, h-current must be present for monensin to decrease period of isolated heart interneurons (bicuculline).”

*2) The h-current can be clearly observed in Figure 5 when membrane potential is hyperpolarized, whereas it is totally absent in the presence of strophanthidin (Figure 5). Any explanations? It is odd that in the presence of strophanthidin there is no sag potential.*

We do not know the mechanisms that would explain the absence of the sag potential when strophanthidin was present, which was observed in five out of five preparations. We can only speculate as follows:

“Because strophanthidin blocks the pump, it increases internal Na^+^ concentrations, which has the effect of changing the equilibrium potential for Na+. However, the equilibrium for K^+^ was maintained by our K^+^-based intracellular electrode. Thus, it is possible that the activation of the h-current led to very little inward current and we may also be observing the activation of an inward rectifier.”

If there’s a preference for this paragraph to be included in the discussion, we would be happy to do so.

[Editors’ note: the author responses to the first round of peer review follow.]

*While all the referees found the paper interesting and potentially important, there was substantial concern about the interpretation of the experiments and the strength of the conclusions that the manuscript makes based on the techniques used. There was also strong agreement among the reviewers and the editors that the paper as presented was incredibly difficult to read and poorly put together especially in terms of the descriptions of the figures in the text and figure legends.*

We reorganized the presentation of our results and figures. We simplified the figure captions by describing only what is actually presented in the figures and restricting discussions on the implications of the figures to the main text.

*The major experimental issue with the paper revolves around the ability to interpret some of the manipulations. The authors would like to be able to conclude that only the contribution of the dynamics of the pump with respect to rhythmic activity is being altered by their conditions. The concern is that manipulating the pump pharmacologically has the fundamental problem that it alters both the dynamics of the pump and its function in stabilizing membrane potentials.*

We put forward a pump model that has no inherent dynamics (Equation 5 in revised manuscript), where “I max pump” is the maximum Na^+^ /K^+^ pump current, [Na]_ih_ is the intracellular Na^+^ concentration for the half-activation of the Na^+^ /K^+^ pump, and [Na]_is_ the sensitivity of the Na^+^ /K^+^ pump to [Na]_i_. This equation is similar to other published pump models (e.g., Gadsby, 1980; Mogul et al., 1990; De Weer et al., 2001). In the text describing the model, we claim and show that the pump current follows the Nai dynamics (now Figure 7). Monensin has the effect of shifting the baseline Na^+^ influx (now Figure 7). Thus, it does not alter the pump dynamics (it has no dynamics) but it does alter the pump current. The pump current does indeed have dynamics and these dynamics are important for the 2 effects it has on electrical activity (now Figure 7). Monensin does change [Na^+^]_i_, which alters the pump current. Although it is true that the off-target effects of monensin are not known in this system, we did perform additional voltage-clamp experiments (now Figure 2) with strophanthidin to show that it alters the pump current.

*In addition, there are concerns about the specificity of the manipulations. Monensin has not been validated in the leech system and the fact that it acts by increasing Na^+^ levels leaves open many possibilities of how it could affect cellular function. Validation by intracellular recording and perhaps showing that its actions are blocked by ouabain would be helpful in establishing its mechanism.*

We performed additional voltage-clamp experiments with strophanthidin to determine if the stimulatory effects of monensin on the pump current can be blocked (now Figure 2). Consistent with previous studies (Lichtshtein et al., 1979; Doebler, 2000; Wang et al., 2012), we found that strophanthidin was able to block the effects of monensin, indicating that monensin does indeed act on the pump current. We also explored further the effects of monensin on bursting activity by performing additional experiments involving intracellular recordings of one of the two oscillator heart interneurons and observing changes in its membrane potential (now Figure 3). We found that monensin decreases the burst period of the intracellularly recorded oscillator heart interneuron, similar to the results from our extracellular experiments. This effect occurred without any significant changes to the neuron’s membrane potential.

*Likewise, the use of zero K^+^ is problematic since this manipulation would affect many processes. Because of the reliance on the extracellular recordings, some of the conclusions are less direct than they need to be, in terms of how membrane potential or K^+^ currents are being affected.*

*The reviewers also did not think that the modeling component of the paper was very useful. Exploring the parameters of the pump dynamics more fully instead of simply showing that the cell could be working as posited would be more rigorous and generally informative.*

The use of K^+^ -free saline to block the pump current is a well-established procedure used in many previous studies (e.g., Garrahan and Glynn, 1967; Carpenter and Alving 1968; Maetz, 1969; Sokolove and Cooke, 1971; Thomas, 1972; Gadsby 1980; Bühler et al., 1991; Lafaire and Schwarz 1986; Catarsi and Brunelli, 1991; Wang et al., 2012). Nevertheless, we do acknowledge the reviewers’ concerns that the results of our K^+^ -free saline experiments could be affected by changes in the equilibrium potential for K^+^. To address these concerns, we performed additional experiments involving multiple lower concentrations of external K^+^ (0.1, 0.4, and 2 mM K^+^). It is only in the K^+^ -free saline experiment that we observed a significantly decreased burst period. In the other experiments involving lower external concentrations of K^+^, the burst period either remained the same or increased significantly. These results strengthen our conclusions that the pump current is critical to normal bursting and that inhibition of the pump speeds up bursting activity, consistent with the inhibitory effects of myomodulin on the pump as revealed by Tobin and Calabrese (2005).

May we add that confidence in strophanthidin (or ouabain) is perhaps misguided. It influences the equilibrium potential of both Na^+^ and K^+^ and causes cell swelling due to osmotic load. In addition, strophanthidin is a poison that can be hard to wash. Overall, K^+^ – free saline is a much gentler, easily reversible, and readily controllable method of blocking the pump than strophanthidin, and we urge the reviewers to try this method on their experimental systems.

We have simplified the presentation of the modelling component by reducing the number of experimental treatments being modeled from eight to three and by removing the sections describing the interaction between the low-threshold slowly-inactivating Ca^2+^ current and the pump current. Thus, we focused solely on how the pump current interacts with the h -current, as described by our new experimental results (now Figure 4). We elaborated further on how this interaction might occur and that the pump current must be dynamic for there to be normal bursting. Simply injecting negative current was not sufficient to produce normal bursting activity. We explored the monensin rate constant in

3 our half-center oscillator model by varying it over an entire range and observing its effects on the burst period (now Figure 7). We found that the burst period decreases nearly linearly with an increase in the rate constant until bursting was halted at rate constants above 2.2 × 10-3 s^-1^. This is consistent with our experimental results showing a slightly shorter burst period under 10 μM monensin saline relative to 1 μM monensin saline (now

Figure 3—figure supplement 1).

*Reviewer #1:*

*The effect of hyperpolarization and its interaction with h-currents has been studied in different contexts, but the potential role for network dynamics has been mostly overlooked. The topic is therefore of potential significance for many systems. However, I found a number of flaws in this study.*

*Although the writing is mostly clear, I find the narrative could be improved. This is mostly due to the sequence in which the data are presented. The paper falls into three sections. First, the effect of monensin on bursting activity and its interaction with h-currents is described. Then follows a set of experiments aimed at providing evidence for the specificity of monensin effects by using intracellular electrophysiology while manipulating the pump in different ways. Last, the modeling work is presented to address the mechanisms underlying the effects on neuronal activity. In my opinion, the flow of this work could be much improved by addressing the cellular actions of monensin first, providing evidence that it is really the pump that is activated. The effect on bursting activity can then be addressed experimentally and theoretically without interruption. In addition, results described in the text and presented in the figures should be in the same sequence, which is not always the case).*

We changed the organization of the revised manuscript and figures using the suggested sequence (cellular effects – bursting activity – model).

*For a number of reasons it is difficult to show that the effects on network activity are really due to selective activation of the pump. For this reason, the authors show a number of approaches to provide parallel lines of evidence. While the voltage-clamp experiments showing that monensin increases outward current without affecting conductance are fairly convincing, other approaches (AHP measurments, zero K^+^, strophanthidin) and are flawed because they allow other interpretations.*

We removed the Results section involving measurements of the afterhyperpolarization. We performed more experiments to address concerns of the K^+^ -free saline experiments (now Figure 6).

*Other than the argument that the computational model can recreate the firing behavior seen in the experiments, it is not critically addressed if monensin effects are really just due to direct action on the interneurons studied. With the interneurons in halfcenter mode, are there other intact inputs? Even in the other set of experiments, doesn't synaptic isolation with bicucullin just mean isolation from each other? There may be good arguments against this being a problem, but for anyone not familiar with the system it isn't evident that the network effects couldn't be due to effects on presynaptic or modulatory neurons.*

We observed the effects of monensin on the membrane potential (now Figure 2) and membrane current (Figure 3) in saline that contained 1.8 mM Mn^2+^, which would have blocked all synaptic and Ca^2+^ currents. We also blocked the h -current with 2 mM Cs+. Thus, the effects of monensin is very unlikely to be attributed to the presynaptic or modulatory neurons. Yes, in this context, synaptic isolation does mean isolation and it has been established that bicuculline does isolate the heart interneurons from all other fast chemical synaptic inputs.

*The computational models replicate the network effects and therefore serve as proof-of-concept, i.e. show that the ionic mechanisms included are sufficient to explain the effects. However, they fall somewhat short in giving a mechanistic explanation of the specific effects. The interactions of pump currents, h-currents, and Ca^2+^ currents are analyzed solely from the model traces replicating the different experimental conditions. The analysis is therefore mostly descriptive and correlative. There is no careful independent manipulation of the critical parameters, and therefore the statements about complex interactions between different currents are mostly hand-waving.*

We have removed the sections describing the interaction between the low-threshold slowly-inactivating Ca^2+^ current and the pump current. We also manipulated the monensin-rate constant— a critical parameter— to demonstrate that there is a near linear relationship between the rate constant and burst period, consistent with our experimental findings (now lines 482-489 and now Figure 7).

*Reviewer #2:*

*Major concerns:*

*1) In this study monensin is used to increase Na^+^ concentrations and in turn enhance Na/K pump current. Experiments have been done to demonstrate that the outward current induced by monensin is the pump current, however more direct evidence is still lacking. This probably can be easily provided by adding ouabain/strophanthidin to the voltage clamped interneurons to block the change of holding current. Or using ouabain/strophanthidin in experiments shown in Figure 5.*

*Although monensin has been used widely to enhance Na/K pump current, it actually unspecifically increase the intracellular Na^+^ concentration, which in turn has effects on various targets, e.g. changing Na^+^ current kinetic and activating Na^+^-dependent K^+^ channels.*

As recommended, we performed additional voltage-clamp experiments, which showed that

the stimulatory effects of monensin on the pump can be blocked by adding

strophanthidin.

*2) Studies have shown that A-type current is also linked to the self-regulation of Na/K pump function, and alter neuron firing, e.g. Pulver and Griffith's paper, which is also cited here. Have the authors considered this possibility in their studies? If this current can be ruled out, it may be worth to mention it in the Discussion.*

In the leech oscillator heart interneurons, the A-type current has very short activation and inactivation kinetics as follows:

a. Activation kinetics:

time constant of ~15 ms at -50 mV and below

time constant of ~5 ms at -20 mV and above

b. Inactivation kinetics:

time constant of ~25 ms at -70 mV and below

time constant of ~35 ms at -40 mV and above

Thus, the activation kinetics of A-type current are much too short (range of milliseconds) to sustain an effect on the burst period (range of seconds), which decreased from 8.2 ± 0.7 s (control saline) to 2.7 ± 0.1 s (monensin saline) (now Figure 4). Moreover, there were no significant changes in the intraburst spike frequency between monensin and control saline treatments (Figure 4), making it difficult to determine if the A-type current plays a role. In addition, extensive modeling experiments in which A– type current was varied show that it has little or no effect on burst period (Hill et al. 2001).

*3) I assume that the dual extracellular recordings with Cs+ and monensin (Figure 1) are always in the sequence of Cs+ and then monensin. Have the authors tried the other way around, adding monensin first then Cs+? This is to rule out the possibility that the sequence may affect the result, and confirm the outcome.*

*The authors compared the burst period, burst duration, and interburst interval of half-center oscillators treated with monensin saline (Figure 2) to those of half-center oscillators treated with Cs+ plus monensin saline (Figure 3), which are unpaired data. If some extra experiments have been done in the sequence suggested, these data can be paired.*

We performed additional experiments in a different order (now Figure 4; control -> monensin -> monensin and Cs+) to control for the effects of the drug sequence being a potential confound in our experimental results. As predicted, we found that adding Cs+ after monensin led to an increase in the interburst interval, which in turn decreased the duty cycle, consistent with our previous experiment that was based on the opposite sequence (Figure 4; Cs+ -> Cs+ and monensin).

Reviewer #3:

*I was greatly relieved when I got to Figure 5 and Figure 6, which presents evidence that the pharmacolocigal agents (10 μM monesin and 2 mM Cs+) used in these experiments are acting via the mechanisms that the authors assume, but by then I had plowed through the detailed analysis of effects of various combinations of monesin, Cs+ and bicuculine on burst period, burst duration, interburst interval, duty cycle and spike frequency, all the while assuming that the authors had not bothered to examine possible alternative pharmacological effects. Some of this problem might be avoided by presenting the results in Figure 5 and Figure 6 before presenting the detailed analysis in Figure 1–Figure 4. This would give the reader more confidence in the interpretation of results.*

We changed the order of presentation by describing the effects of monensin on the membrane potential and membrane current first, followed by the effects of monensin on bursting activity.

*The authors also need to cite evidence that 2 mM Cs+ is acting via its effect on the h-current, rather than some other K^+^ current. I am hoping that the effects of 2 mM Cs+ have previously been substantiated in these neurons, because this key assumption is not sufficiently supported by the experiments in this paper.*

The effects of Cs+ on the h-current have indeed been previously substantiated in these neurons. To make that point clear, we added the following sentence to the Results section:

“Angstadt and Calabrese (1989) found that the h-current in leech oscillator heart interneurons can be blocked by 2-4 mM Cs+.”

*Any way that the authors can find to reorganize the results to make them more easily digested would be helpful.*

We hope you will find the new rearrangement in the revised manuscript to be satisfactory.

*Reviewer #4:*

*Essentially the authors wish to show that the Na/K pump is dynamically activated during bursting in leech such that the pump interacts with Ih to help determine oscillator period. This would make a nice and important result.*

*That said, it almost feels as if there are 2-3 papers here, each of which needs more fleshing out, which are not done justice by putting them together. The theory section seems too fragile to be more than a "it could work like this". Having a model to say "it could work like this" isn't necessarily bad, but is counterproductive unless it is relatively robust to the natural variation in the membrane conductances in the system, which it may or may not be.*

We have reduced the modeling section by removing the section describing the interaction between the low-threshold slowly-inactivating Ca^2+^ current and the pump current. We also manipulated the monensin-rate constant— a critical parameter— to demonstrate that there is a near linear relationship between the rate constant and burst period (now lines 482-489 and now Figure 7), consistent with our experimental findings showing a decrease in the burst period under 10 μM relative to 1 μM monensin (now Figure 3—figure supplement 1).

*The issues in the paper that confound its simple interpretation:*

*a) The authors use 10uM monensin to putatively activate the pump by increasing the intracellular Na concentration. This is an established drug (but a higher concentration than often used), with known pharmacological properties in other systems, but we are not given any indications that the authors have validated its use in leech, except to show that its effects are similar to direct Na injection. It is not totally obvious what the validation could be, but minimally, some kind of dose response using intracellular recordings to see that the change in membrane potential is a graded function of drug concentration. And that the effects are counteracted by oubain, in a dose-dependent manner? Would that be helpful? And were these high concentrations necessary?*

We did perform intracellular recordings whereby we observed membrane potential of an oscillator heart interneuron that had been isolated with Mn^2+^ and Cs+ (now Figure 1). In the absence of rhythmic activity, we found that monensin hyperpolarized the membrane potential of the isolated neuron. We also performed an additional voltage-clamp experiment whereby we used strophanthidin to block the monensin-enhanced pump current (now Figure 2). Finally, we performed combined intracellular and extracellular recordings of the oscillator heart interneurons in half-center configuration and found that the burst period decreased significantly, without any significant change in the base potential (now Figure 3).

We performed additional experiments with monensin at a lower concentration of 1 uM. At this concentration, it took longer for the maximal effects of monensin to be observed.

Thus, we opted to use the concentration of 10 μM because its maximal effects can be observed sooner (now Figure 3—figure supplement 1). Moreover, the 10 μM is widely used 12 in many other systems (e.g., Lichtshtein et al., 1979; Tsuchida and Otomo, 1990; Busciglio et al., 1993; Itoh et al., 2000; Lamy and Chatton, 2011; Wang et al., 2012; Zhang et al., 2015), which allows us to compare our results to those from other studies. Some studies (e.g., Satoh and Tsuchida, 1999) have even used higher concentrations of monensin (e.g., 20-30 μM monensin) to stimulate the Na^+^ /K^+^ pump.

*b) Using zero extracellular K is a strange thing to do, as that means the K equilibrium potential is now ill-defined. It would be far better to do normal, 1/2, 1/10 K, so that it would be possible to interpret the results of low K on the outward currents.*

The use of K^+^ -free saline to block the effects of the pump is well-established (Garrahan and Glynn, 1967; Carpenter and Alving 1968; Maetz, 1969; Ritchie, 1971; Sokolove and Cooke, 1971; Thomas, 1972; Gadsby 1980; Bühler et al., 1991; Lafaire and Schwarz 1986; Catarsi and Brunelli, 1991; Wang et al., 2012). Nevertheless, we take seriously the concern that the issue of the K^+^ equilibrium potential being ill-defined. As recommended, we performed additional experiments with lower external concentrations of K^+^ (0.1 mM, 0.4 mM, and 2 mM). In contrast to our K^+^ -free experiment, the burst period either remained the same or actually increased when the neurons were bathed in saline with lower external concentrations of K^+^. These experiments further support our K^+^ -free experiments as well as the previous myomodulin experiments from our lab (Tobin and Calabrese, 2005), which showed that inhibition of the pump speeds up bursting activity as indicated by a shorter burst period (now Figure 6).

May we add that confidence in ouabain (or strophanthidin) can be misguided. It influences the equilibrium potential of both Na^+^ and K^+^ and causes cell swelling due to osmotic load.

*In addition, ouabain is a poison that can be hard to wash. Overall K^+^ -free saline is a much gentler, reversible, and easily controllable method of blocking the pump than strophanthidin, and we strongly suggest that this method be tested on the reviewer’s experimental system.*

*c) The modeling section is unsatisfying in that it tries to do too much and therefore it does too little. For example, it would be interesting to see the effect of changes of the monensin rate on the behavior of the isolated single neuron or the half-center. As it stands, it sounds like the monensin rate was changed for different simulations? Did I misunderstand this? And it would be great to see the effect of different moninsen rates on neurons with different Ih's. Did I miss this? If the model is as sensitive as it seems that it might be, what does this mean for the biological system? And wouldn't that have necessitated matching with different drug concentrations? This is all confusing.*

We have reduced the number of experimental treatments being modeled from eight to three. We also included a figure panel showing the manipulation of the monensin rate constant in our half-center oscillator model (now Figure 7). Based on that manipulation, the burst period decreases incrementally with each successive increase in the monensin rate constant (now lines 482-489). This is consistent with our experimental results whereby the burst period in 10 μM monensin was lower than the one under 1 μM monensin treatment (now Figure 3—figure supplement 1).

Yes, the monensin rate constant was changed for different simulations to provide the best fit to our experimental data. We found that the half-center oscillator model with the blocked h -current was more sensitive and therefore had a rate constant that was one order of magnitude lower than the regular half-center oscillator model.